# Zinc finger protein Zfp335 is required for the formation of the naïve T cell compartment

Brenda Y Han[1], Shuang Wu[1], Chuan-Sheng Foo[2], Robert M Horton[1], Craig N Jenne[1‡], Susan R Watson[1†], Belinda Whittle[3], Chris C Goodnow[4], Jason G Cyster[1*]

[1]Department of Microbiology and Immunology, Howard Hughes Medical Institute, University of California, San Francisco, San Francisco, United States; [2]Department of Computer Science, Stanford University, Stanford, United States; [3]Australian Phenomics Facility, John Curtin School of Medical Research, Australian National University, Canberra, Australia; [4]Department of Immunology, John Curtin School of Medical Research, Australian National University, Canberra, Australia

**Abstract** The generation of naïve T lymphocytes is critical for immune function yet the mechanisms governing their maturation remain incompletely understood. We have identified a mouse mutant, *bloto*, that harbors a hypomorphic mutation in the zinc finger protein Zfp335. *Zfp335*^bloto/bloto mice exhibit a naïve T cell deficiency due to an intrinsic developmental defect that begins to manifest in the thymus and continues into the periphery, affecting T cells that have recently undergone thymic egress. The effects of Zfp335^bloto are multigenic and cannot be attributed to altered thymic selection, proliferation or Bcl2-dependent survival. Zfp335 binds to promoter regions via a consensus motif, and its target genes are enriched in categories related to protein metabolism, mitochondrial function, and transcriptional regulation. Restoring the expression of one target, Ankle2, partially rescues T cell maturation. These findings identify Zfp335 as a transcription factor and essential regulator of late-stage intrathymic and post-thymic T cell maturation.

*For correspondence: jason.
cyster@ucsf.edu

†Deceased

Present address: ‡Snyder Institute for Chronic Diseases, University of Calgary, Calgary, Canada

Competing interests: The authors declare that no competing interests exist.

## Introduction

In order to mount effective adaptive responses against a diverse range of pathogens and antigens, the immune system has to generate sufficient numbers of mature peripheral T cells that express functional T cell receptors (TCRs). T cell development is a complex and highly regulated process that involves multiple stages of selection and maturation, both within the thymus and after thymic export. In the thymus, productive rearrangement of the TCR β-chain in CD4⁻CD8⁻ double negative (DN) thymocytes drives progression to the CD4⁺CD8⁺ double positive (DP) stage (*Starr et al., 2003*). After rearrangement of the TCR α-chain, DP thymocytes express mature TCRs which are then used to survey self-peptide/MHC complexes presented by specialized epithelial cells in the thymic cortex (cTECs) (*Klein et al., 2014*). A small percentage of DP thymocytes receive positively selecting TCR signals which promote their survival, in part through upregulation of IL-7Rα (*Sinclair et al., 2011*). Positively selected thymocytes become committed to either the CD4 or CD8 single-positive (SP) lineage and migrate to the thymic medulla, where they undergo further negative selection mediated by interactions with antigen-presenting cells such as dendritic cells (DCs) or AIRE-dependent medullary thymic epithelial cells (mTECs) (*Hogquist et al., 2005*; *Klein et al., 2014*), during which thymocytes expressing self-reactive TCRs either undergo apoptosis or are diverted to alternative fates, such as becoming regulatory T cells (Tregs) (*Stritesky et al., 2012*).

As SP thymocytes undergo maturation, expression of the surface marker CD24 is decreased while CD62L expression is upregulated. As such, SP thymocytes may be further divided into two phenotypically

**eLife digest** To defend our bodies against a variety of foreign microbes, our immune system makes cells called T cells that can identify these invaders and help to destroy them. There are several types of T cells that play different roles in the immune response: some activate other immune cells, while others destroy cells that have been infected by viruses or other pathogens.

T cells develop in a specialized organ called the thymus, where they go through a rigorous selection process before being released as mature T cells into the rest of the body. This selection process includes eliminating individual T cells that are found to be sub-standard, perhaps because they might mistake our own cells for enemy cells. However, many of the details of the later stages of T cell development are not fully understood.

Han et al. have now found that a protein called Zfp335 that is involved in the production of mature T cells. Mice carrying a mutation in the gene that makes this protein have fewer mature T cells than normal mice. Han et al. also reveal that Zfp335 is a transcription factor that can control whether or not other genes are expressed as proteins, and further show that one of these proteins, Ankle2, has an important role in the production of mature T cells.

A next step in the work is to define exactly how Zfp335 controls the expression of these genes. It will also be important to determine whether mutations in Zfp335 contribute to human T-cell immunodeficiency.

distinct populations, often referred to as semi-mature ($CD62L^{lo}CD24^{hi}$) and mature ($CD62L^{hi}CD24^{lo}$). These phenotypic changes are associated with an important functional difference: semi-mature SP thymocytes are susceptible to apoptosis upon TCR stimulation, whereas mature SP thymocytes are not and respond instead by proliferating (*Sprent and Kishimoto, 2002*; *Weinreich and Hogquist, 2008*). In addition, only mature SP thymocytes upregulate sphingosine-1-phosphate receptor (S1PR1), which is required for egress from the thymus (*Matloubian et al., 2004*; *Weinreich and Hogquist, 2008*). The process of SP thymocyte maturation, from entry into the SP compartment to thymic egress, has been estimated to take 4–5 days (*McCaughtry et al., 2007*). New T cells, also referred to as recent thymic emigrants (RTEs), undergo a phase of post-thymic phenotypic and functional maturation before they are incorporated into the long-lived peripheral naïve T cell pool (*Fink, 2012*).

The transition from the semi-mature to mature stage of SP thymocyte development is marked by numerous changes in gene expression. Some of these changes, including the upregulation of S1PR1 and CD62L, are mediated by the transcription factor KLF2 (*Carlson et al., 2006*). In the periphery, the process of post-thymic maturation is also associated with transcriptional changes, though on a smaller scale (*Mingueneau et al., 2013*). For instance, an increase in IL-7Rα expression during this period has been shown to promote recent thymic emigrant (RTE) survival (*Silva et al., 2014*). Proper regulation of the transcriptional program underlying late-stage SP thymocyte and post-thymic T cell maturation is thus critically important for the establishment of a normal naïve T cell compartment. Multiple genes involved in NF-κB signaling have been reported to be required for the development of mature T cells (*Guerin et al., 2002*; *Schmidt-Supprian et al., 2003*; *Sato et al., 2005*; *Zhang and He, 2005*; *Liu et al., 2006*; *Wan et al., 2006*; *Silva et al., 2014*), primarily through mechanisms related to TCR signaling and protection from apoptosis. In addition, roles for the transcriptional repressor Nkap (*Hsu et al., 2011*) and chromatin remodeling factor Bptf (*Landry et al., 2011*) have been identified in recent years. However, the transcriptional regulators controlling these stages of T cell maturation remain largely unknown.

The C2H2 zinc finger family constitutes the largest class of transcription factors in mammalian genomes, and many key transcriptional regulators in immune cell development, such as Ikaros and Plzf, contain multiple C2H2 zinc fingers (*Brayer and Segal, 2008*). The C2H2 zinc finger fold is classically recognized to be a DNA-binding domain (*Wolfe et al., 2000*; *Iuchi, 2005*), although it may also participate in interactions with RNA (*Brown, 2005*) or other proteins (*Brayer and Segal, 2008*).

In this study, we identify a C2H2 zinc finger protein, Zfp335, as an essential regulator of T cell maturation. Zfp335, also known as NIF-1 (*Mahajan et al., 2002*; *Garapaty et al., 2009*), is ubiquitously expressed and is essential for early development, with homozygous deletion resulting in embryonic lethality at E7.5 (*Yang et al., 2012*). Here, we report that an ENU-induced mutant allele of Zfp335

results in defective accumulation of naïve T cells, largely as a consequence of impaired maturation in SP thymocytes and RTEs. We show that this maturation defect is independent of thymic selection or effects on proliferation, but is associated with reduced viability. We identify a set of Zfp335 target genes in thymocytes and present evidence that decreased Zfp335 occupancy at a subset of these targets alters gene expression in mutant thymocytes. Taken together, our findings provide evidence that Zfp335 functions as a transcription factor and key regulator of a transcriptional program required for T cell maturation.

## Results

### The ENU mouse mutant *bloto* has a deficiency in peripheral T cells

As part of an *N*-ethyl-*N*-nitrosourea (ENU) mutagenesis screen (*Nelms and Goodnow, 2001*) for lymphocyte phenotypes, we identified a variant C57BL/6 mouse pedigree with decreased frequencies of CD4$^+$ and CD8$^+$ T cells in peripheral blood (*Figure 1A*), which we named *bloto* (blood T cells low; allele henceforth designated *blt*). This trait was fully penetrant and occurred at a frequency consistent with recessive inheritance. Homozygotes were viable, fertile and displayed no gross external abnormalities.

Further characterization of the *bloto* mutant revealed a strong reduction in overall T cell frequencies in secondary lymphoid organs, especially in the CD62L$^{hi}$CD44$^{lo}$ naïve T cell population (*Figure 1B*). Analysis of T cell development in the thymus revealed no significant decrease in frequencies or numbers of CD4$^-$CD8$^-$ DN or CD4$^+$CD8$^+$ DP thymocytes of *blt/blt* mice relative to heterozygous controls (*Figure 1C,D*). However, *blt/blt* mice had slightly lower SP thymocyte frequencies, and subgating on semi-mature (CD62L$^{lo}$CD24$^{hi}$) and mature (CD62L$^{hi}$CD24$^{lo}$) SP thymocytes showed significant underrepresentation of the mature subset (*Figure 1C*), with an approximately twofold decrease in the numbers of both CD4 and CD8 mature SP thymocytes (*Figure 1D*). In comparison, CD4$^+$ and CD8$^+$ naïve T cell numbers in the spleen were reduced about fivefold to eightfold (*Figure 1E*), suggesting both a thymic and peripheral component to the *bloto* T cell developmental defect. In mixed bone marrow chimeras, lower percentages of *blt/blt* as compared to wild-type cells were observed in the SP thymocyte and naïve T cell populations, demonstrating that the T cell phenotype is cell-intrinsic and recapitulating the progressive developmental defect seen in intact mice (*Figure 1F*). The decrease in *blt/blt* T cells in mixed chimeras was comparable to that in intact mice (*Figure 1—figure supplement 1A*), which indicates the lack of a competitive or rescue effect by wild-type cells. The *bloto* phenotype is a fully recessive trait with no evidence for haploinsufficiency or a dominant negative effect, since heterozygous mice exhibited no decrease in naïve T cells compared to wild-type controls (*Figure 1— figure supplement 2A*), and *blt/+* T cells did not decline relative to wild-type cells even in a competitive mixed chimeric setting (*Figure 1—figure supplement 2B*).

Despite the strong defect in naïve T cells, we noted little difference in the number of T cells with an effector/memory phenotype (CD62L$^{lo}$CD44$^{hi}$) (*Figure 1—figure supplement 3A*). This is likely due to homeostatic expansion of surviving cells in T cell-deficient *blt/blt* mice, as this increase in memory relative to naïve T cells was not observed in mixed chimeras in which the effects of T cell lymphopenia were alleviated by the presence of wild-type cells (data not shown). Non-conventional αβT cell lineages, such as Foxp3$^+$ regulatory T cells and iNKTs, were also affected (*Figure 1—figure supplement 3B*), but not to a greater degree than conventional CD4$^+$ and CD8$^+$ T cells. However, there were no deficiencies in other major lymphocyte lineages such as NK cells (*Figure 1F*; *Figure 1—figure supplement 3C*), γδT cells, and B cells (*Figure 1—figure supplement 3C*), suggesting that the *bloto* mutation has a selective effect on αβT cells.

### Identification of a missense mutation in *Zfp335*

To identify the causative genetic lesion, the peripheral blood T cell deficiency was used to map the mutation in an F2 intercross to a genomic interval between 163.16 and 165.88 Mb on chromosome 2 (*Figure 2—figure supplement 1A*). Whole-exome sequencing of DNA from an affected mouse identified a single novel single-nucleotide variant within the interval of interest: a C to T missense mutation in exon 21 of *Zfp335* (*Figure 2A*). This results in the replacement of a positively charged Arg residue at position 1092 (henceforth referred to as R1092) by Trp, a bulky non-polar amino acid.

Zfp335 is a 1337-amino acid protein containing 13 predicted C2H2 zinc finger domains (*Figure 2B*). Its role as a transcriptional regulator in neurogenesis and neuronal differentiation has recently been described (*Yang et al., 2012*), but any immunological function has thus far been unknown. The R1092W

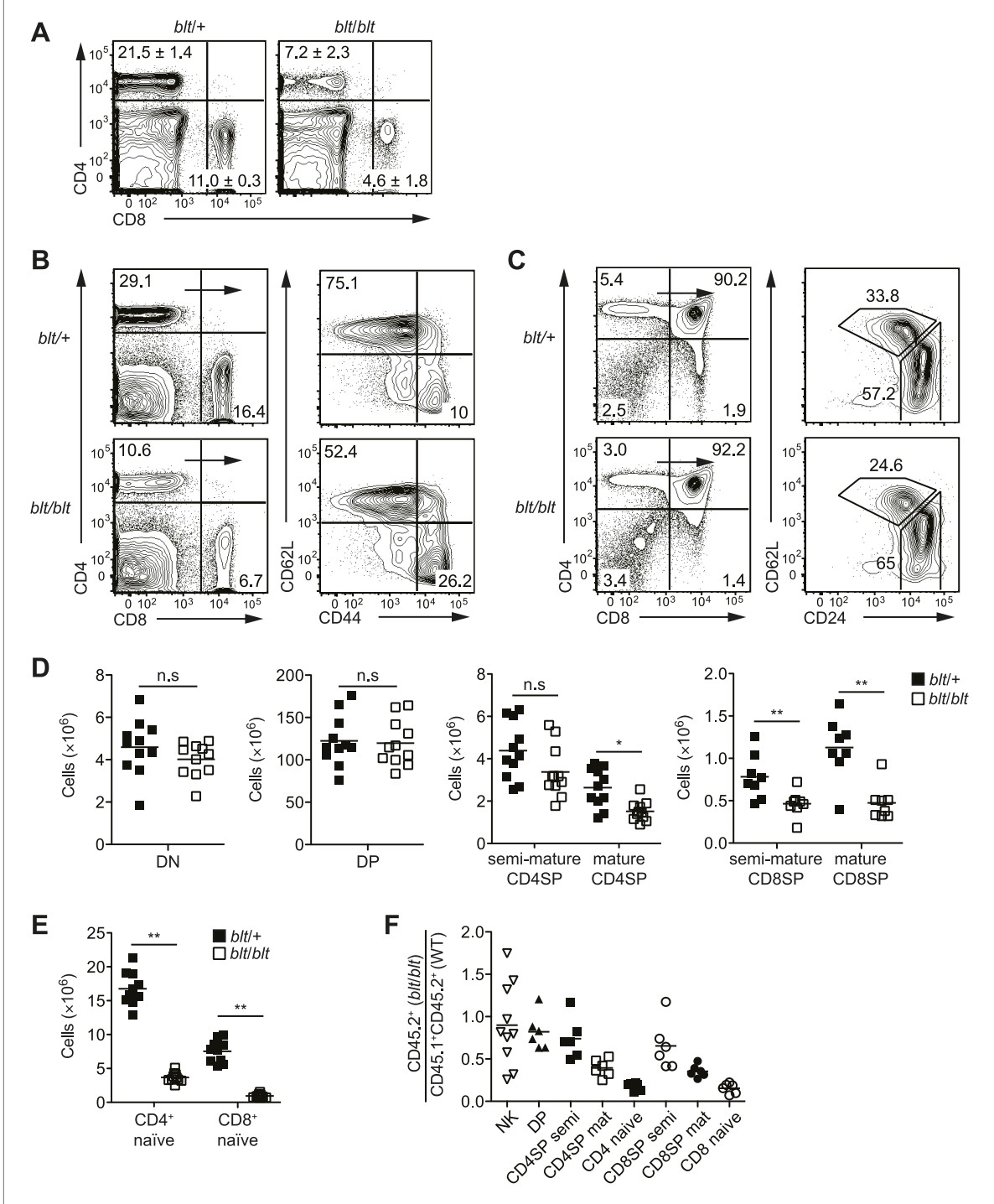

**Figure 1**. Identification of an ENU mouse mutant with a cell-intrinsic deficiency in peripheral T cells. (**A**) Frequency of CD4+ and CD8+ T cells in peripheral blood of 8-week-old heterozygous (*blt/+*) or homozygous (*blt/blt*) mice as detected by flow cytometry. Numbers in quadrants indicate mean frequencies ± s.d. (*n* = 3 mice per genotype). (**B**) Frequency of total splenic CD4+ and CD8+ T cells (left); percentage of CD4+ T cells with a naïve (CD62L^hi^CD44^lo^) or effector (CD62L^lo^CD44^hi^) phenotype (right). (**C**) Frequency of major thymocyte subsets (left); proportion of semi-mature (CD62L^lo^CD24^hi^) and mature (CD62L^hi^CD24^lo^) subsets within the CD4SP thymocyte population (right). (**D**) Absolute number of DN, DP, CD4SP, and CD8SP thymocytes in *blt/+* vs *blt/blt* mice. Semi-mature and mature CD4SP thymocytes were gated as in (**C**). CD8SP thymocytes were gated as follows: semi-mature (TCRβ^hi^CD62L^lo^CD24^int^), mature (TCRβ^hi^CD62L^hi^CD24^lo^). Mature CD4SP and CD8SP thymocytes are reduced in numbers by approximately 1.8- and 2.3-fold, respectively. (**E**) Quantification of CD4+ and CD8+ naïve T cells in the spleen, gated as in (**B**), showing a 4.6-fold and 7.8-fold decrease in CD4+ and CD8+ naïve T cells,

*Figure 1. Continued on next page*

*Figure 1. Continued*

respectively. (**F**) Ratio of *blt/blt* (CD45.2$^+$) vs wild-type (CD45.1$^+$CD45.2$^+$) cells for splenic NK cells (NK1.1$^+$TCRβ$^-$), DP, semi-mature (semi) and mature (mat) SP thymocytes and naïve splenic T cells from lethally irradiated WT CD45.1$^+$ hosts reconstituted with a 1:1 mix of *blt/blt* and WT bone marrow cells. Data in (**B**) and (**C**) are representative of seven to eight independent experiments with matched *blt/+* and *blt/blt* littermates and are summarized in (**D**) and (**E**). Mice were analyzed at 8 to 10 weeks of age (**A–E**) or 8 to 12 weeks post-reconstitution (**F**). Each symbol represents an individual mouse; small horizontal lines indicate the mean; n.s, not significant; *p < 0.05 and **p < 0.01 (two-tailed Mann–Whitney test).

The following figure supplements are available for figure 1:

**Figure supplement 1**. Similar relative decrease in blt/blt T cells in mixed chimeras vs intact mice, indicating the lack of a competitive or rescue effect by WT cells.

**Figure supplement 2**. Mice heterozygous for the bloto mutation do not exhibit a T cell phenotype.

**Figure supplement 3**. blt/blt mice exhibit a selective defect in αβ T cells.

mutation falls within the 12th zinc finger (ZF12) near the C-terminus (*Figure 2B*), at a position that is highly conserved across vertebrate evolution (*Figure 2C*). Homology modeling places R1092 in the ZF12 α-helix at position +6 (*Figure 2—figure supplement 2A,B*), one of the canonical positions mediating DNA base recognition by C2H2 zinc fingers (*Wolfe et al., 2000*). The presence of a TNEKP linker between ZF12 and ZF13 (*Figure 2—figure supplement 2A*) and its similarity to the conserved TGEKP linker, a key structural feature of DNA-binding C2H2 zinc fingers (*Wolfe et al., 2000*), further hint at the possibility that R1092 may play a direct role in DNA binding by Zfp335.

Zfp335 transcript levels were not decreased in *blt/blt* thymocytes and T cells; in fact, a slight increase was observed relative to *blt/+* controls, particularly in the most mature subsets (*Figure 2D*). Western blotting analysis of thymocytes showed no reduction in the amount of Zfp335 protein (*Figure 2E*), indicating that the R1092W mutation had no adverse effect on protein expression or stability. Confocal imaging of sorted mature SP thymocytes showed that both wild-type and mutant Zfp335 localized to the nucleus, forming punctate foci within regions of euchromatin (*Figure 2F*). No detectable differences in subnuclear distribution were observed. These data suggest that the *bloto* mutation is hypomorphic rather than null, as it results in normal levels of stable protein that can localize appropriately to the nucleus but has impaired function due to the selective disruption of ZF12.

An in vivo gene complementation test was carried out by retroviral transduction of wild-type Zfp335 into *blt/blt* bone marrow for hematopoietic reconstitution of irradiated hosts. The T cell development block was strongly reversed in *blt/blt* cells transduced with wild-type Zfp335 but not control vector (*Figure 2G,H*), hence establishing that Zfp335$^{R1092W}$ was the causative mutation. Overexpression of Zfp335$^{R1092W}$ yielded an intermediate rescue effect (*Figure 2H*), suggesting that supraphysiological protein expression may partially compensate for impaired function caused by a hypomorphic mutation. Interestingly, we observed that transduction frequencies for Zfp335$^{WT}$ in DP thymocytes (*Figure 2H*) or non-T lymphocytes (data not shown) were typically low (<10%) compared to transduction frequencies achieved with Zfp335$^{R1092W}$ or other genes, suggesting that overexpression of Zfp335 may have an inhibitory effect on early hematopoiesis, leading to poorer reconstitution of transduced progenitors.

## *blt/blt* mice have defects in SP thymocyte maturation and homeostasis of recent thymic emigrants

To further characterize the block in intrathymic development, we examined thymocyte populations by continuous in vivo bromodeoxyuridine (BrdU) labeling, where the percentage of BrdU$^+$ cells indicates population turnover. After 4 days of BrdU administration, when comparing *blt/blt* mice to heterozygous controls, the mature SP population showed a decrease in turnover, whereas thymocyte subsets from earlier stages of development labeled with similar kinetics (*Figure 3A*; *Figure 3—figure supplement 1A*). Genome-wide transcriptome analysis of sorted mature CD4SP thymocytes revealed a gene expression profile consistent with impaired SP thymocyte maturation; by showing, for instance, decreased expression of genes known to be upregulated during maturation (*Teng et al., 2011*) (*Figure 3B*). By comparing staining intensities of various surface markers associated with SP maturation (e.g., CD24, CD62L) in pre-gated mature SP subsets, we also observed a trend towards a less mature surface phenotype (data not shown). Taken together, these data suggest that *blt/blt* mice have decreased efficiency of

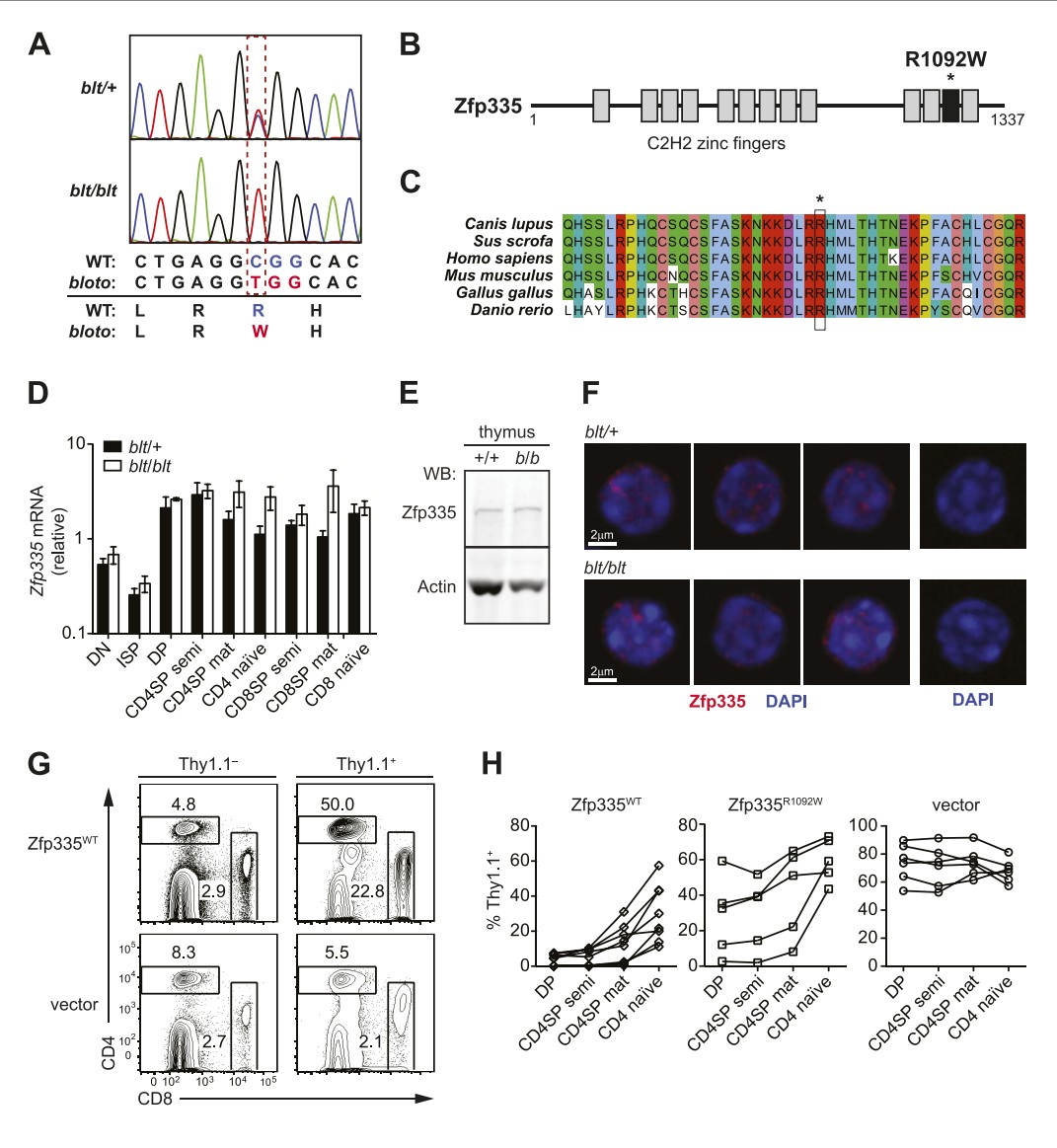

**Figure 2**. Identification of causative missense mutation within a C2H2 zinc finger of Zfp335. (**A**) Sequence trace analysis of the mutated codon in homozygous (*blt*/*blt*) compared to heterogyzous (*blt*/+) mice, showing an Arg-to-Trp substitution at position 1092. (**B**) Linear schematic of the 13 C2H2 zinc finger (ZF) domains (shaded boxes) in Zfp335. Asterisk indicates the R1092W *bloto* mutation in ZF12 (black box). Diagram drawn to approximate scale. (**C**) Multiple sequence alignment of predicted Zfp335 orthologs from dog (*Canis lupus*), pig (*Sus scrofa*), human (*Homo sapiens*), mouse (*Mus musculus*), chicken (*Gallus gallus*), and zebrafish (*Danio rerio*). Asterisk indicates Arg residue affected by *bloto* mutation. Amino acids are colored according to their physicochemical proper-ties. (**D**) Quantitative RT-PCR analysis of *Zfp335* mRNA from indicated FACS-purified thymocyte subsets and naïve T cells ($n$ = 3–4 mice, mean ± s.d. for biological replicates); ISP, immature CD8+ thymocytes identified by tlack of TCRβ expression; results are presented relative to expression of *Hprt*. (**E**) Western blot for Zfp335 protein in the thymocytes from wild-type (+/+) and homozygous mutant (*b*/*b*) mice, with actin as loading control. (**F**) Immunofluorescence analysis of Zfp335 nuclear localization in mature CD4SP thymocytes; nucleus counterstained with DAPI. (right) Secondary antibody-only negative staining control. Scale bar: 2 μm. (**G**) Frequency of CD4+ and CD8+ T cells differentiating from *blt*/*blt* hematopoietic stem cells transduced (Thy1.1+) with either wild-type Zfp335 (Zfp335WT) or control MSCV-IRES-Thy1.1 vector, compared to non-transduced (Thy1.1−) cells from the same mouse, 8 to 10 weeks after reconstitution of irradiated hosts. (**H**) Transduced (Thy1.1+) cells as a percentage of indicated thymocyte and T cell subsets from irradiation chimeras that had received bone marrow retrovirally transduced with

*Figure 2. Continued on next page*

*Figure 2. Continued*

WT Zfp335, *bloto* Zfp335 or control vector. Data points are connected by a separate line for individual mice. Data are representative of three independent experiments.

The following figure supplements are available for figure 2:

**Figure supplement 1**. Linkage mapping of bloto mutation to a 2.72 Mb region on chromosome 2 containing Zfp335.

**Figure supplement 2**. Protein sequence analysis and structural modeling of mutated C2H2 zinc finger in Zfp335.

entry into the mature SP compartment and progression through the final maturation stages within the mature SP thymocyte population.

As described earlier, *blt/blt* mice have a more severe defect in the accumulation of naïve T cells compared to that of mature SP thymocytes, suggesting that Zfp335$^{R1092W}$-induced dysregulation extends to events in the periphery following thymic export. To assess the impact of Zfp335$^{R1092W}$ on naïve T cell homeostasis, we adoptively transferred *blt/blt* peripheral T cells together with wild-type controls into congenic lymphoreplete hosts and assessed their relative maintenance over 7 days. Surprisingly, *blt/blt* T cells persisted just as well as control wild-type T cells (***Figure 3—figure supplement 1B***), suggesting that Zfp335$^{R1092W}$ does not significantly impair the survival of the bulk naïve T cell population. This led us to hypothesize that the *bloto* defect may be largely confined to new T cells that have recently left the thymus, otherwise known as recent thymic emigrants (RTEs). In adult mice, these cells comprise a relatively small percentage of total naïve T cells and may therefore not be significantly represented in measurements involving the bulk naïve T cell population. We initially assessed the ability of SP thymocytes to survive once introduced into the periphery, in essence behaving as surrogate RTEs. Significantly, *blt/blt* cells decayed more rapidly than co-transferred controls, hinting that RTE survival may be impaired in *blt/blt* mice (***Figure 3—figure supplement 1B***).

In order to study the RTE population in *blt/blt* mice in greater detail, we crossed the *blt/blt* mutant to the Rag1-GFP reporter line (***Kuwata et al., 1999***). In these reporter mice, GFP signal intensity is inversely proportional to time spent in the periphery (***Boursalian et al., 2004***), allowing for the identification of RTEs as GFP$^+$ cells within the naïve T cell population. We found that the GFP$^{hi}$ subset was significantly overrepresented in *blt/blt* naïve T cells and was skewed towards cells with the highest GFP signal intensities that have most recently exited the thymus, consistent with a partial block in post-thymic naïve T cell maturation (***Figure 3—figure supplement 1C***). To directly assess RTE maintenance in vivo, a test population of either Rag1-GFP *blt/+* or Rag1-GFP *blt/blt* peripheral T cells was mixed with control non-fluorescent T cells and injected i.v. into congenic lymphoreplete hosts. Consistent with previous experiments (***Figure 3—figure supplement 1B***), there was no significant decline in relative numbers of total *blt/blt* naïve T cells recovered from the spleen 1, 3, and 5 days post-transfer (***Figure 3C***). However, the *blt/blt* Rag1-GFP$^+$ population declined more rapidly than *blt/+* Rag1-GFP$^+$ controls, particularly within the first day of adoptive transfer (***Figure 3C***). Interestingly, in contrast to the spleen, we noted a small decrease in the maintenance of total *blt/blt* naïve T cells, relative to *blt/+* cells, that were recovered from peripheral lymph nodes. Similarly, the lymph nodes exhibited a larger decline in *blt/blt* Rag1-GFP$^+$ relative to *blt/+* Rag1-GFP$^+$ T cells (***Figure 3D***). These data suggest that some of the apparent decline in *blt/blt* RTEs (***Figure 3D***) may be due to less efficient short-term accumulation in lymph nodes. Nonetheless, the fact that we observe in both spleen and lymph nodes a higher rate of decay in the Rag1-GFP$^+$ population as compared to the overall effect on the bulk naïve T cell pool strongly suggests that *blt/blt* RTEs survive less well than control RTEs.

In terms of absolute numbers, GFP$^{hi}$ RTEs in *blt/blt* mice were reduced by a magnitude largely matching that of GFP$^-$ mature naïve cells (***Figure 3—figure supplement 1D***), suggesting that most of the drop-off in cell numbers may have occurred at the earliest stages of RTE maturation. To test this hypothesis, we treated mice with FTY720, a potent inhibitor of thymic egress (***Matloubian et al., 2004***). This strategy ensured that new T cells, which would have otherwise been exported into the periphery, were trapped in the thymus where their accumulation could be measured. After 4 days of thymic egress blockade, *blt/blt* mice showed greatly impaired accumulation of SP cells with a mature CD62L$^{hi}$CD24$^{lo}$ phenotype (***Figure 3E***), suggesting that most of the losses in RTEs take place within

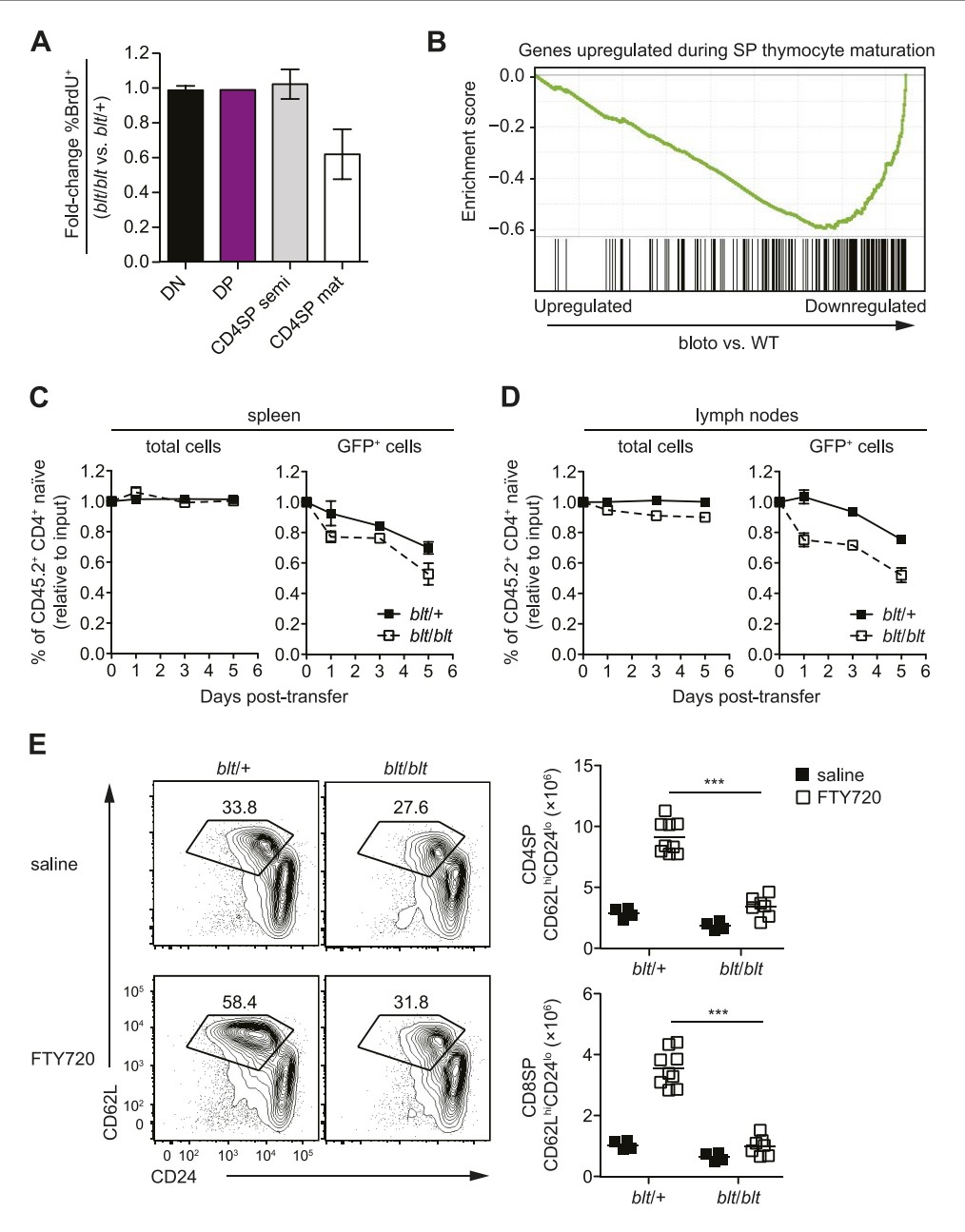

**Figure 3**. Zfp335[R1092W]-induced T cell dysregulation affects mainly mature SP thymocytes and recent thymic emigrants. (**A**) Percentage of BrdU+ cells in indicated thymocyte populations from *blt/blt* mice relative to *blt/+* controls after 4 days of continuous BrdU labeling (mean ± s.d., *n* = 4). (**B**) Gene set enrichment analysis (GSEA) analysis of gene expression data from *blt/blt* vs WT mature CD4SP thymocytes showing significant negative correlation with genes known to be upregulated during SP thymocyte maturation (MSigDB gene set: GSE30083). (**C** and **D**) Input-normalized fraction of total (left panel) or GFP+ (right panel) Rag1-GFP *blt/+* and Rag1-GFP *blt/blt* cells within the total CD45.2+ CD4+ naïve donor population recovered from recipient spleens (**C**) and peripheral lymph nodes (**D**) at indicated time points after co–transfer with control CD45.2+ *blt/+* cells. (**E**) Flow cytometry analysis of CD4SP cells in the thymus of *blt/blt* mice and *blt/+* controls after 4 days of FTY720 or saline treatment. Percentage of cells in mature SP (CD62LhiCD24lo) gate shown. Results are quantified (right) for CD4SP and CD8SP thymocytes. ***p < 0.001 (two-tailed Mann–Whitney test), data pooled from four independent experiments.

The following figure supplement is available for figure 3:

**Figure supplement 1**. Impaired late-stage SP thymocyte development and early post-thymic peripheral T cell maturation in blt/blt mice.

a short period after they enter the periphery. These data also indicate that the cell loss is due to intrinsic defects in the maturing T cells and is not dependent on their location in a particular lymphoid compartment.

## Intact thymic selection in *blt/blt* mice

In mice with a polyclonal TCR repertoire, we observed normal frequencies of positively selected DP thymocytes with a CD69$^{hi}$TCRβ$^{int}$ phenotype (*Figure 4—figure supplement 1A*), with no differences in CD5 surface expression on post-selection thymocytes (*Figure 4—figure supplement 1B*), suggesting that positive selection is not strongly impaired by the *bloto* mutation. Because compensatory TCR rearrangements may mask a potential positive selection defect, we examined thymic development in *blt/blt* mice expressing the class II MHC-restricted OTII TCR transgene, in which impaired positive selection would be expected to cause a dramatic reduction in CD4SP thymocytes. However, this was not observed in OTII *blt/blt* mice, which had only slightly lower CD4SP frequencies compared to controls and a fold reduction in CD4SP numbers (*Figure 4A*) similar to that seen in polyclonal *blt/blt* mice (*Figure 1D*).

Using the OTII/RIP-mOVA model of AIRE-dependent clonal deletion (*Anderson et al., 2005*), we found no evidence that *blt/blt* mice had altered negative selection (*Figure 4B*). Consistent with this conclusion, mRNA expression of Nur77, a key proapoptotic regulator induced during negative selection (*Baldwin and Hogquist, 2007*), was not elevated in *blt/blt* semi-mature SP thymocytes (*Figure 4—figure supplement 1D*). Hence, our data indicate that altered thymic selection does not contribute to the T cell deficiency in *blt/blt* mice.

## The *bloto* T cell deficiency is not due to defects in Bcl2-dependent survival, IL-7Rα expression or proliferation

The findings in *Figure 3C,D* and *Figure 3—figure supplement 1B* suggested decreased survival may at least in part explain why *blt/blt* T lymphocytes fail to accumulate normally. Consistent with this notion, *blt/blt* mature SP thymocytes showed a greater loss of viability in vitro over time compared to *blt/+* controls, while a lesser effect was seen for *blt/blt* semi-mature SP cells (*Figure 4C*). Annexin V staining and measurement of active caspase 3 in freshly isolated thymocytes did not reveal differences between *blt/blt* and *blt/+* mice (data not shown), likely because cells in the earliest stages of apoptosis are efficiently cleared in vivo (*Surh and Sprent, 1994*). To examine whether Bcl2-regulated apoptotic pathways were involved in promoting the loss of *blt/blt* T cells, we crossed *blt/blt* mice to a transgenic line expressing human BCL2 under control of the *Lck* promoter (*Sentman et al., 1991*). Overexpression of BCL2 failed to rescue the defect in peripheral naïve *blt/blt* T cells (*Figure 4D*), indicating that it is not due to reduced Bcl2-dependent survival. In addition, no differences in the expression of Bcl2 family pro- and anti-apoptotic genes were observed in *blt/blt* mature SP thymocytes (*Figure 4—figure supplement 2A*).

IL-7 is a critical regulator of naïve T cell homeostasis (*Surh and Sprent, 2008*); in particular, IL-7Rα expression is induced in new T cells and is required for their survival and integration into the peripheral pool (*Silva et al., 2014*). However, no significant differences in IL-7Rα expression were detected in *blt/blt* thymocytes and T cells, either at the transcript level (*Figure 4—figure supplement 2B*) or by surface receptor staining (*Figure 4—figure supplement 2C*). Furthermore, survival of *blt/blt* thymocytes and naïve T cells in the presence of IL-7 in vitro was comparable to that of co-cultured wild-type cells (data not shown), suggesting that the T cell defect is not due to the loss of IL-7R function.

Lastly, Zfp335$^{R1092W}$ had no significant effect on the fraction of thymocytes in cell cycle, as shown by short-term BrdU labeling (*Figure 4—figure supplement 3A*) and DNA content analysis (*Figure 4—figure supplement 3B*). Similarly, *blt/blt* naïve T cells were able to expand at a normal rate following TCR stimulation in vitro (*Figure 4—figure supplement 3C*), demonstrating that *blt/blt* T cells are not defective in their ability to undergo proliferation.

## Zfp335 binds to active gene promoters in thymocytes

Zfp335 has recently been shown to bind a variety of gene promoters in mouse embryonic brain (*Yang et al., 2012*), but its genome-wide binding characteristics and targets relevant to its function in T cell development remain to be defined. In addition, because our structure-function predictions (*Figure 2—figure supplement 2*) led us to hypothesize a DNA-binding role for the mutated Arg, we wished to know if the R1092W mutation disrupted the ability of Zfp335 to bind to its targets in vivo.

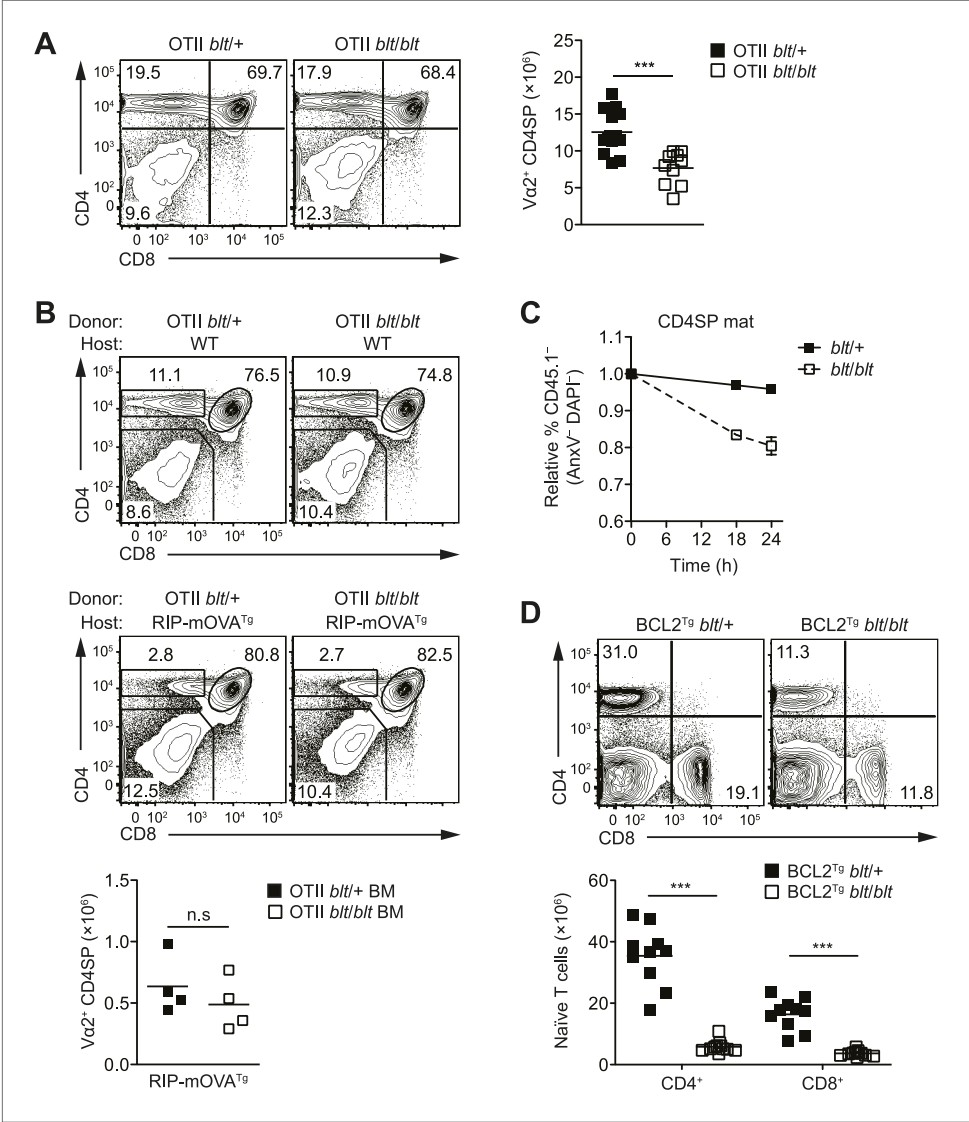

**Figure 4**. The T cell maturation defect in blt/blt mice is not caused by altered thymic selection or Bcl2-dependent survival. (**A**) Flow cytometry analysis of major thymocyte populations from OTII TCR transgenic blt/+ and blt/blt mice, gated on total live thymocytes (left); quantification of OTII TCR-expressing Vα2+ CD4SP thymocytes from OTII blt/+ (n = 14) and OTII blt/blt (n = 14) mice (right). (**B**) Frequency of thymocyte subsets in lethally irradiated WT B6 (top left) or RIP-mOVATg (bottom left) recipients reconstituted with T cell-depleted bone marrow from OTII blt/+ and OTII blt/blt mice; quantification of Vα2+ CD4SP thymocytes from indicated RIP-mOVATg chimeric mice (right). (**C**) In vitro viability of sorted CD45.1− blt/+ vs blt/blt semi-mature (left) and mature (right) CD4SP thymocytes co-cultured with CD45.1+ WT CD4SP thymocytes. Live cells were pre-gated as annexin V− DAPI− and percentage of CD45.1− cells was normalized to input. (**D**) Frequency of CD4+ and CD8+ T cells in the spleen of blt/+ vs blt/blt mice expressing a human BCL2 transgene (BCL2Tg) under the control of the proximal Lck promoter (left); quantification of naïve T cells from BCL2Tg blt/+ (n = 10), and BCL2Tg blt/blt (n = 10) mice (right). Data representative of eight (**A**), three (**B**), two (**C**), and six (**D**) independent experiments; n.s, not significant, ***p < 0.001 (two-tailed Mann–Whitney test).

The following figure supplements are available for figure 4:

**Figure supplement 1**. blt/blt mice exhibit intact positive and negative selection in the thymus.

**Figure supplement 2**. Normal expression of IL-7 receptor and Bcl2 family members.

**Figure supplement 3**. blt/blt naïve T cells proliferate normally in response to TCR stimulation in vitro and show no significant reduction in cycling of mature SP thymocytes.

To address these questions, we performed ChIP-seq analysis of Zfp335 binding sites in total thymocytes isolated from wild-type and *blt/blt* mice. ChIP-seq data sets were generated using two separate polyclonal antibodies and are referred to as 'ChIP-C' and 'ChIP-N' respectively. Using data obtained from wild-type thymocytes, peak calling with a *q*-value threshold of <0.05 identified 157 Zfp335-binding regions in the vicinity of 177 genes. Genome browser inspection of ChIP-seq signal tracks confirmed that these peaks, although fairly limited in number, represented regions of significantly enriched binding intensity. Genome-wide, Zfp335 peaks were strongly enriched in gene promoters (**Figure 5A**) and located upstream of transcriptional start sites (TSS) (**Figure 5B**). Zfp335-bound regions were associated with high levels of H3K4me3, a hallmark of active gene promoters, and low levels of the enhancer-associated modification H3K27ac and the repressive chromatin mark H3K27me3 (**Figure 5C**), consistent with Zfp335 functioning primarily as a regulator of promoter-dependent gene transcription. Zfp335 target genes were enriched for functional categories representing a diverse range of biological processes, including protein synthesis and metabolism, mitochondrial function, cell cycle regulation, RNA processing, and transcriptional regulation (**Figure 5D**; **Supplementary file 4**).

## Decreased Zfp335 binding at a subset of target genes in *blt/blt* thymocytes

With the same *q*-value cutoff (<0.05) that yielded a total of 157 binding events for wild-type thymocytes, we detected 141 peaks in *blt/blt* thymocytes (**Supplementary file 2,3**). By visual inspection of ChIP-seq data on a genome browser, we determined that of the 28 peaks detected in wild-type but not *blt/blt* thymocytes, 22 showed a convincing loss of binding while the rest were false positives due to noise at low-confidence peaks. Of the nine peaks that were called for Zfp335$^{R1092W}$ but not Zfp335$^{WT}$,

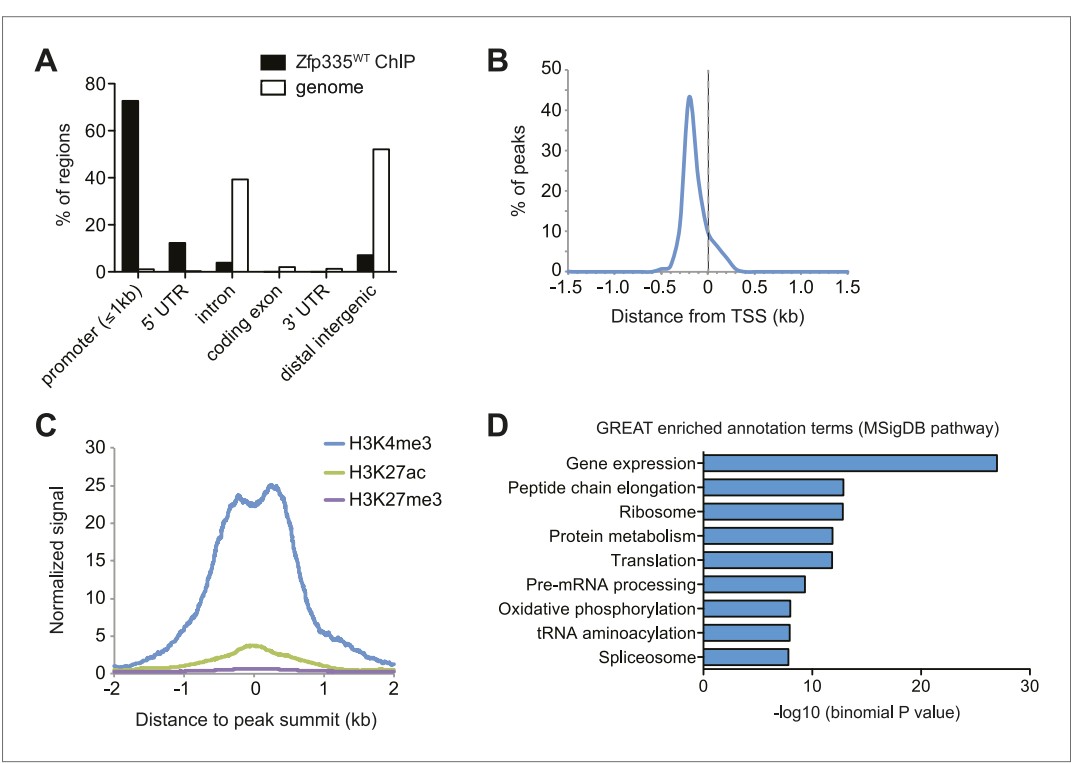

**Figure 5**. Genome-wide analysis of Zfp335 binding sites in wild-type thymocytes based on ChIP-seq using an antibody against the C-terminus of Zfp335. (**A**) Genomic feature annotation of Zfp335 peaks reveals strong enrichment in promoter regions (≤1 kb upstream of TSS) and 5′ UTRs relative to genomic background. (**B**) Average profile of peak center distances from nearest RefSeq TSS for 141 Zfp335 peaks located within ±1.5 kb of a TSS, showing a positional preference for binding upstream of the TSS. (**C**) Average density of H3K4me3 (blue), H3K27ac (green) and H3K27me3 (purple) marks for a region from −2 kb to +2 kb relative to Zfp335 peak summits, based on ENCODE histone modification ChIP-seq data for murine whole thymus. (**D**) Gene ontology analysis of genes associated with Zfp335 binding sites using GREAT. Top enriched annotation terms in the MSigDB pathway ontology are shown.

four were not true binding events, but rather signal artifacts arising from repeat regions, while the remaining five were low-confidence peaks. This strongly suggests that the *bloto* mutation does not lead to the gain of novel binding sites, which is consistent with our hypothesis that it is a loss-of-function hypomorph.

Interestingly, we did not observe a global decrease in Zfp335 binding intensities across all target sites. Reduced binding was detected for a subset of target genes: *Ankle2*, *Nme6*, and *Mrps5* are shown as representative examples (*Figure 6A*). However, at other target sites, such as *Rbbp5*, *Polr2e*, and *Pes1*, Zfp335 binding did not appear to be significantly impaired (*Figure 6B*). Only nine Zfp335-bound regions showed decreases in ChIP-seq peak intensities in *blt/blt* thymocytes that were robust enough to be detected across both sets of ChIP-C and ChIP-N replicates (*Figure 6C*). We performed ChIP-qPCR to validate a selection of target genes that we had identified as differentially bound by Zfp335$^{R1092W}$, vs targets that showed normal binding, and found the ChIP-qPCR data to be in agreement with the ChIP-seq-based assessment (*Figure 6D*). Moreover, ChIP-qPCR analysis of sorted CD4SP thymocytes yielded similar results to the analysis with total thymocytes (*Figure 6—figure supplement 1A*).

To understand how reduced Zfp335 binding at a subset of direct targets in *blt/blt* thymocytes could be biologically significant, we integrated our ChIP-seq data with gene expression profiles that we had obtained for *blt/blt* and wild-type mature CD4SP thymocytes. Of the ten target genes listed in *Figure 6C* as having significantly reduced Zfp335$^{R1092W}$ binding, only three (*Ankle2*, *Nme6*, *Cnpy2*) were down-regulated in expression by more than twofold (*Figure 6E*). Since our ChIP-seq results were derived from total thymocytes, of which only a small percentage are mature CD4SP, one caveat was that we might have missed gene expression changes in the larger population, so we sorted DP thymocytes (>90% of total) and tested them by RT-qPCR. Relative to *blt/+* controls, *blt/blt* DP thymocytes had no detectable differences in the expression of *Mrps5* and *Rabggtb*, even though these genes exhibit strongly reduced Zfp335 binding in *blt/blt* thymocytes (*Figure 6—figure supplement 1B*). This is not an unexpected finding given that transcription factors are known to bind sites which they do not functionally regulate (*Smale, 2014*). Nonetheless, we did observe deregulated expression of many Zfp335 target genes in our CD4SP array data, including many which did not exhibit reduced Zfp335 occupancy according to our stringent criteria (*Figure 6—figure supplement 1C*). For example, although *Wdr47* failed to meet these criteria because it showed decreased Zfp335$^{R1092W}$ binding in the ChIP-C but not ChIP-N data set (*Figure 6—figure supplement 1D,E*), its expression was nonetheless significantly downregulated in *blt/blt* thymocytes (*Figure 6E*, *Figure 6—figure supplement 1C,F*). With the exception of a handful of strongly downregulated targets, gene expression changes were usually mild. Most deregulated target genes exhibited decreased expression, though a few were modestly upregulated (*Figure 6—figure supplement 1C*). Interestingly, this small group of upregulated targets include *Zfp335* itself, which we confirmed by RT-qPCR (*Figure 2D*), suggesting that Zfp335 participates in an autoregulatory negative feedback loop. It is probably reasonable to assume that many of these differentially expressed genes do in fact have reduced Zfp335$^{R1092W}$ binding at their promoters, which we failed to detect owing to limitations in ChIP-seq sensitivity and/or the inability of our approach to identify differential binding events that are not evident at the whole population level because they occur exclusively in mature CD4SP thymocytes.

## Identification of a novel consensus motif for Zfp335

As a member of the C2H2 zinc finger protein family, it is highly likely that Zfp335 interacts with its genomic targets through direct binding to a specific DNA sequence motif. Using a set of high-confidence peaks from our Zfp335$^{WT}$ ChIP-seq data, we performed de novo motif analysis and identified a novel 22 bp bipartite motif consisting of two conserved elements separated by a variable spacer (*Figure 7A*). The positional distribution of this putative consensus motif was unimodal and located near the centers of Zfp335 peaks (*Figure 7B*), consistent with the hypothesis that it is the direct DNA-binding motif. Motif sites found within Zfp335 peaks showed a distinct DNase I genomic footprint (*Figure 7C*) and strong sequence conservation across evolution (*Figure 7D*) compared with motif sites outside peaks, further suggesting that it is a functional DNA-binding motif for Zfp335.

To determine if Zfp335 was able to bind to this DNA sequence in vitro, we performed gel shift assays with labeled oligonucleotide probe containing the predicted consensus motif (*Figure 7E*) and 293T cell nuclear extracts ectopically expressing Zfp335 protein. We found that Zfp335 formed a gel shift complex with the labeled probe (*Figure 7F*). This complex increased in abundance proportional

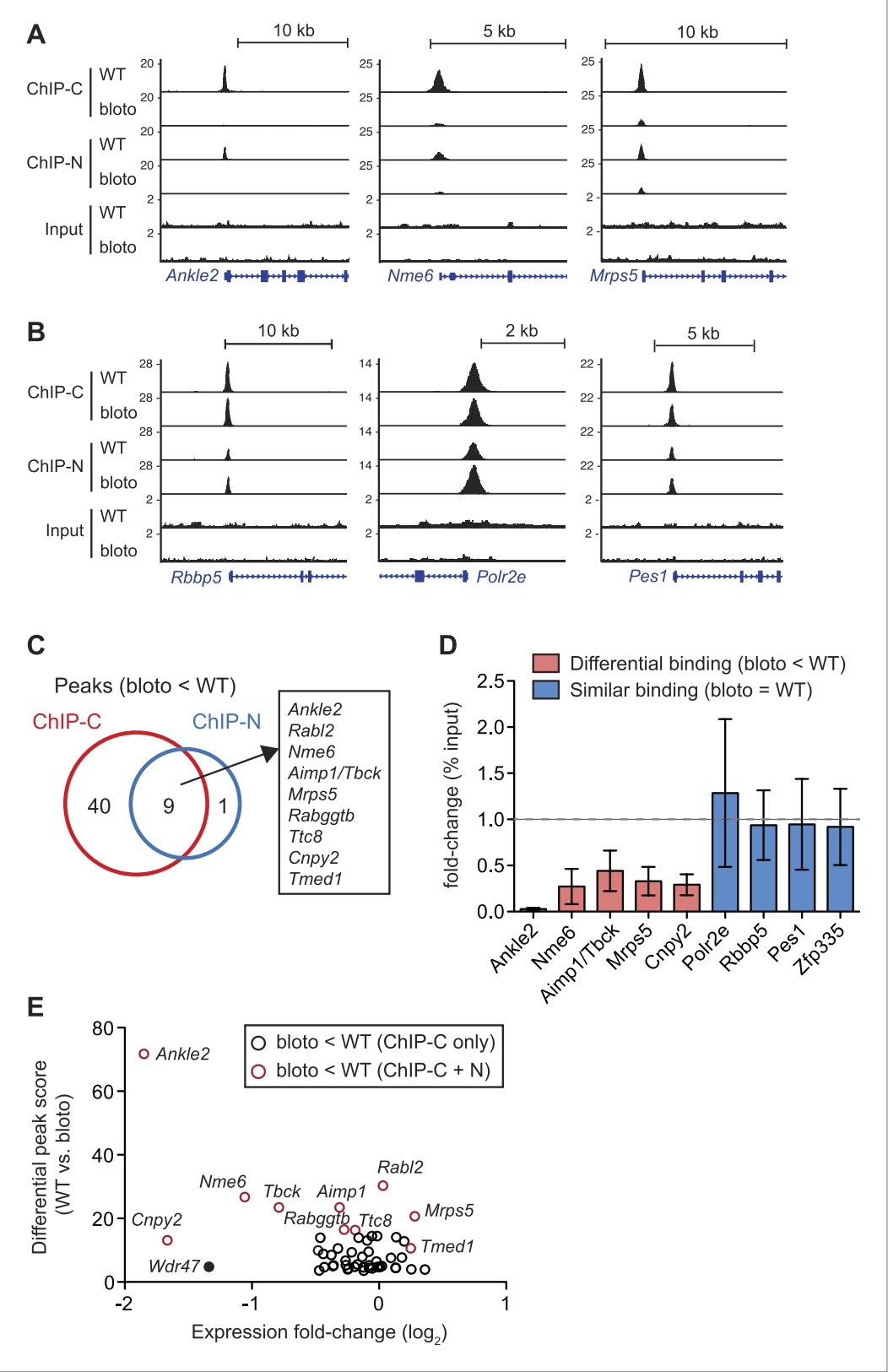

**Figure 6**. Decreased Zfp335 binding in blt/blt thymocytes is detected for a subset of target genes. (**A**) Signal tracks showing Zfp335 occupancy at three target genes (*Ankle2, Nme6, Mrps5*) for which significantly decreased binding in *blt/blt* relative to WT thymocytes is observed using both the C-terminus-specific antibody (ChIP-C) and the
*Figure 6. Continued on next page*

*Figure 6. Continued*

N-terminus-specific antibody (ChIP-N). Vertical axis, fragment pileup per million reads (normalized to library sequencing depth). Input, sequencing of input genomic DNA (background control). (**B**) Signal tracks showing Zfp335 occupancy at three target genes (*Rbbp5, Polr2e, Pes1*) for which no reduction in binding is detected in *blt/blt* thymocytes. (**C**) Identification of nine putative differentially bound target sites and their ten associated genes from the intersection of differential peaks (bloto < WT) called for both ChIP-C and ChIP-N data sets. We consider *Aimp1* and *Tbck* to be associated with a single peak as they share a bidirectional promoter. (**D**) ChIP-qPCR analysis of Zfp335 binding at selected targets to validate ChIP-seq-based assessment of differential binding in *blt/blt* thymocytes. ChIP enrichment was calculated as percent input; results are presented as the fold-change in ChIP enrichment for *blt/blt* vs WT (mean ± s.d., *n* = 3 for three independent experiments). (**E**) Relative Zfp335 binding and gene expression changes for target genes associated with the ChIP-C set of differentially bound regions: horizontal axis, expression fold-change ($\log_2$) values from microarray analysis of *blt/blt* vs WT mature CD4SP thymocytes; vertical axis, score reflecting likelihood that Zfp335 binding is significantly enriched in WT relative to *blt/blt* thymocytes. Red circles, target genes identified as differentially bound in both ChIP-C and ChIP-N data sets (*Figure 6C*); black circles, target genes associated with reduced Zfp335 binding in the ChIP-C but not ChIP-N data set. *Wdr47* is highlighted (filled black circle) as a target gene that was identified as differentially bound only in the ChIP-C data set (*Figure 6—figure supplement 1C,D*) but showed significantly downregulated expression in *blt/blt* thymocytes (*Figure 6—figure supplement 1B,E*).

The following figure supplement is available for figure 6:

**Figure supplement 1**. Analysis of correlation between changes in Zfp335 binding and gene expression in blt/blt thymocytes.

---

to the amount of Zfp335 protein and was eliminated by competition with excess unlabeled probe, demonstrating that it is sequence-specific and Zfp335-dependent (*Figure 7F*). Competition experiments with unlabeled probes containing mutations of conserved nucleotides (*Figure 7E*) showed that Zfp335 binding was abolished when both DNA elements of the bipartite motif were mutated, whereas mutations in either the first or second element had an intermediate effect, indicating that both parts of the consensus motif are required for full Zfp335 binding (*Figure 7G*). We noted a stronger effect on competition upon mutation of the first element compared to the second, suggesting that it makes a greater contribution to Zfp335 binding (*Figure 7G*). We also performed gel shift assays with an oligonucleotide containing the previously reported Zfp335 recognition motif (*Yang et al., 2012*), but failed to detect binding above the negative control (Ikaros binding site) (*Figure 7—figure supplement 1*). Re-analysis of the embryonic brain ChIP-Seq data published by Yang et al. according to the parameters we applied to our data set (see 'Materials and methods') identified the motif shown in *Figure 7A*. These observations suggest that the discrepancy between our and the previously reported motif arises from differences in motif finding strategy. Importantly, the identification of a common motif in these distinct data sets strongly suggests that this sequence element is recognized by Zfp335 in a diversity of cell types.

## Ankle2 dysregulation by Zfp335^R1092W contributes to the T cell maturation defect

To understand which direct targets of Zfp335 were functionally relevant to the T cell maturation defect in *blt/blt* mice, we focused our efforts on a set of genes for which we had clear evidence of reduced Zfp335 occupancy and mRNA expression, the most prominent of which was *Ankle2* (ankyrin repeat and LEM domain-containing protein 2). Zfp335 binding to the *Ankle2* promoter was effectively abolished in *blt/blt* thymocytes (*Figure 6A,D*); this was accompanied by decreased transcript expression across multiple stages of T cell development (*Figure 8A*) and consequently, a virtual absence of Ankle2 protein (*Figure 8B*). Exogenous expression of Zfp335 significantly increased *Ankle2* mRNA levels in *blt/blt* T cells (*Figure 8C*), providing evidence that Zfp335 is both necessary and sufficient for Ankle2 expression.

We were able to partially reverse the T cell maturation defect in the periphery by exogenously expressing Ankle2 in Rag1-GFP *blt/blt* cells (*Figure 8D,E*), as shown by the overall increase in representation of Ankle2-transduced (Thy1.1 reporter+) cells within the more mature Rag1-GFPlo naïve T cell subset compared to the less mature Rag1-GFPhi subset. However, overexpression of Ankle2 had a weaker effect compared to that achieved by Zfp335 (*Figure 8E*), consistent with the idea that *Ankle2*,

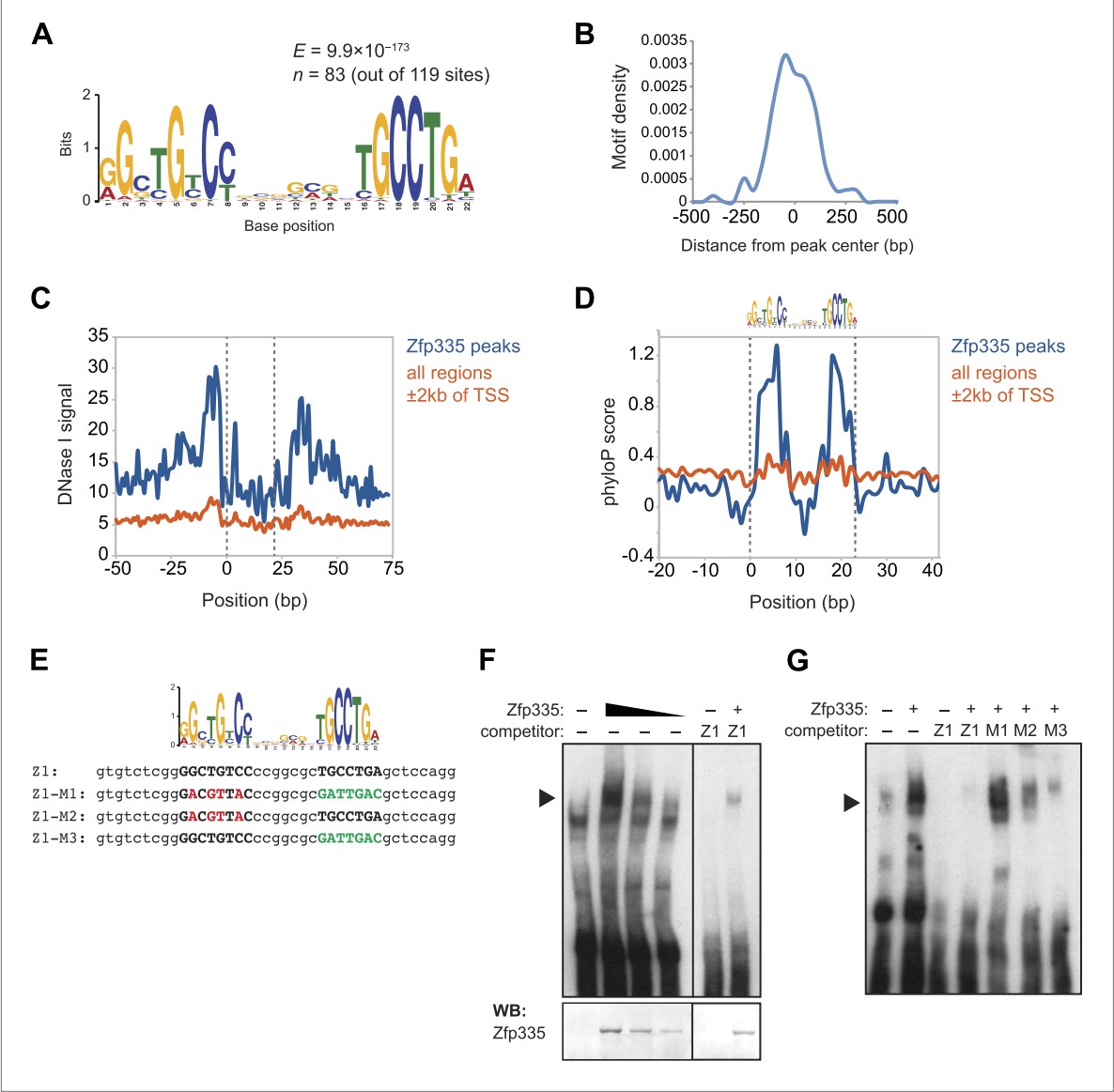

**Figure 7**. Identification of a novel DNA motif bound by Zfp335. (**A**) Sequence motif identified by de novo motif search of WT Zfp335 ChIP-seq peaks. (**B**) Density histogram showing localization of motif relative to Zfp335 peaks. (**C**) DNase I genomic footprinting analysis of motif sites in Zfp335 ChIP-seq peaks (blue) compared with motif sites in all regions ±2 kb of TSS (orange), using ENCODE DGF data for whole thymus. The 22 bp motif is marked on both sides by dashed lines. (**D**) Sequence conservation (phyloP) profiles around Zfp335 motif sites within ChIP-seq peaks (blue) vs sites in all regions ±2 kb of TSS (orange). (**E**) Sequences of oligonucleotide probes used in (**F**) and (**G**). Z1 probe sequence was derived from Zfp335 binding site at the *Zfp335* promoter and contains the primary consensus motif (capitalized, bold letters). For probes M1–M3, the first half (red), second half (green), or both parts of the consensus motif are mutated as shown. (**F**) Gel shift assay demonstrating sequence-specific binding of Zfp335 protein to labeled Z1 probe. Nuclear extracts from 293T cells transfected with control (−) or FLAG-Zfp335 (+) expression vectors were used. Signal from Zfp335-specific complexes (black arrowhead) is eliminated with an excess of unlabeled Z1 competitor oligo. Relative amounts of total Zfp335 protein verified by Western blot (bottom panel). Data are representative of three independent experiments. (**G**) Zfp335 binding to labeled Z1 probe in the presence of competition from unlabeled mutant oligos Z1–M1, Z1–M2, and Z1–M3. Signal intensity inversely correlates with ability of mutant probe to bind Zfp335: M1 is least able to bind, followed by M2, then M3. Black arrowhead, Zfp335 complex. Data are representative of three independent experiments.

The following figure supplement is available for figure 7:

**Figure supplement 1**. Further EMSA characterization of Zfp335-binding motif.

though important, is but one of several downstream targets that are required for Zfp335-dependent T cell maturation. In addition to *Ankle2*, we tested five other Zfp335 target genes for their ability to rescue the T cell maturation defect in *blt/blt* bone marrow chimeras, but did not detect a significant effect for any of these genes (*Figure 8—figure supplement 1*).

## Discussion

In this study, we identify a novel role for Zfp335 as an essential regulator of T cell maturation. By analyzing mice with a hypomorphic missense mutation in a C2H2 zinc finger of Zfp335, we reveal a selective defect in the accumulation of naïve T cells resulting from a maturation block in SP thymocytes and recent thymic emigrants. In line with another recent study (*Yang et al., 2012*), we have shown that Zfp335 regulates transcription by binding to promoters of target genes. We have identified a set of direct targets in thymocytes and provide evidence that Zfp335 occupancy at a small subset of target sites was significantly decreased in mutant T cells. Zfp335 target genes were enriched in categories related to protein metabolism, mitochondrial function, and transcriptional regulation. In addition, we identified a new DNA recognition motif that is bound by Zfp335. Taken together, our findings suggest that Zfp335 acts as a novel transcription factor required for regulating expression of multiple genes required for late stage naïve T cell maturation.

We provide evidence that Zfp335 regulates T cell maturation in part by promoting *Ankle2* transcription. A study in *Caenorhabditis elegans* and HeLa cells showed a role for Ankle2 in nuclear envelope reassembly by promoting dephosphorylation of BAF during mitotic exit (*Asencio et al., 2012*). As we found no significant defects in proliferation (*Figure 4—figure supplement 3*) and did not detect nuclear envelope abnormalities in *blt/blt* mature SP thymocytes by immunofluorescence microscopy (data not shown), it is likely that the requirement for Ankle2 is dependent on some other as-yet-unknown function of this protein. To our knowledge, this is the first time an in vivo role for Ankle2 has been reported, and a more detailed understanding of its function and mechanism of action in T cells will be an important subject for future studies.

The T cell deficiency in *blt/blt* mice does not appear to be a consequence of defects in thymic selection, proliferation, or IL-7Rα expression, but is associated with reduced viability of mature SP thymocytes and recent thymic emigrants. Our finding that BCL2 overexpression failed to rescue the relative deficiency in *blt/blt* T cells, together with the unaltered expression of Bcl2 family genes (*Figure 4—figure supplement 2A*) or other well-defined pro-apoptotic genes such as members of the death receptor family (data not shown), suggests Zfp335 has an indirect and likely multigenic pro-survival influence. Although our data do not provide support for altered thymic selection being an explanation for the reduced T cell numbers in *blt/blt* mice, we cannot exclude the possibility that the selection of some T cell specificities is affected by the Zfp335 mutation and future deep sequencing studies will be needed to fully address this issue. The intact proliferation of *blt/blt* T cells contrasts with the defective proliferation of human lymphoblastic cells and neuronal stem cells carrying a H1111R mutation in ZFP335 (ZNF335) (*Yang et al., 2012*). We suspect that this difference is a consequence of the almost complete loss of protein caused by the H1111R mutation, compared to the more subtle influence of the R1092W mutation studied here.

Naïve T cell deficiencies have been observed in several mouse lines with deletions in genes related to NF-κB signaling, such as RelB (*Guerin et al., 2002*), NEMO (*Schmidt-Supprian et al., 2003*), c-FLIP (*Zhang and He, 2005*), TAK1 (*Sato et al., 2005*; *Liu et al., 2006*; *Wan et al., 2006*), and IKK2 (*Schmidt-Supprian et al., 2003*; *Silva et al., 2014*). However, the expression of these genes was not altered in *blt/blt* T lymphocytes and we did not detect enrichment for NF-κB targets in the gene expression analysis (data not shown). Two other genes known to be required for T cell maturation are the transcriptional repressor Nkap (*Hsu et al., 2011*) and Bptf, a component of the ISWI-containing chromatin remodeling complex NURF (*Landry et al., 2011*). Similar to what we observed in *blt/blt* mice, the maturation block in Nkap- and Bptf-deficient T cells was not caused by altered thymic selection and could not be rescued by Bcl2 overexpression. However, neither Nkap nor Bptf showed significant expression changes in *blt/blt* T lymphocytes. Moreover, we did not find a strong correlation between the expression profiles of *blt/blt* and Bptf-deficient SP thymocytes (unpublished observation). Nonetheless, this does not exclude the possibility of a partial overlap between genes regulated by Bptf and Zfp335—one potential mechanism could be that Bptf-dependent nucleosome repositioning is required for efficient Zfp335 binding to some of its target sites.

A small number of studies have shown that autophagy plays a role in mature T cell survival (*Pua et al., 2007*; *He et al., 2012*; *Parekh et al., 2013*), possibly by ensuring the clearance of excess mitochondria

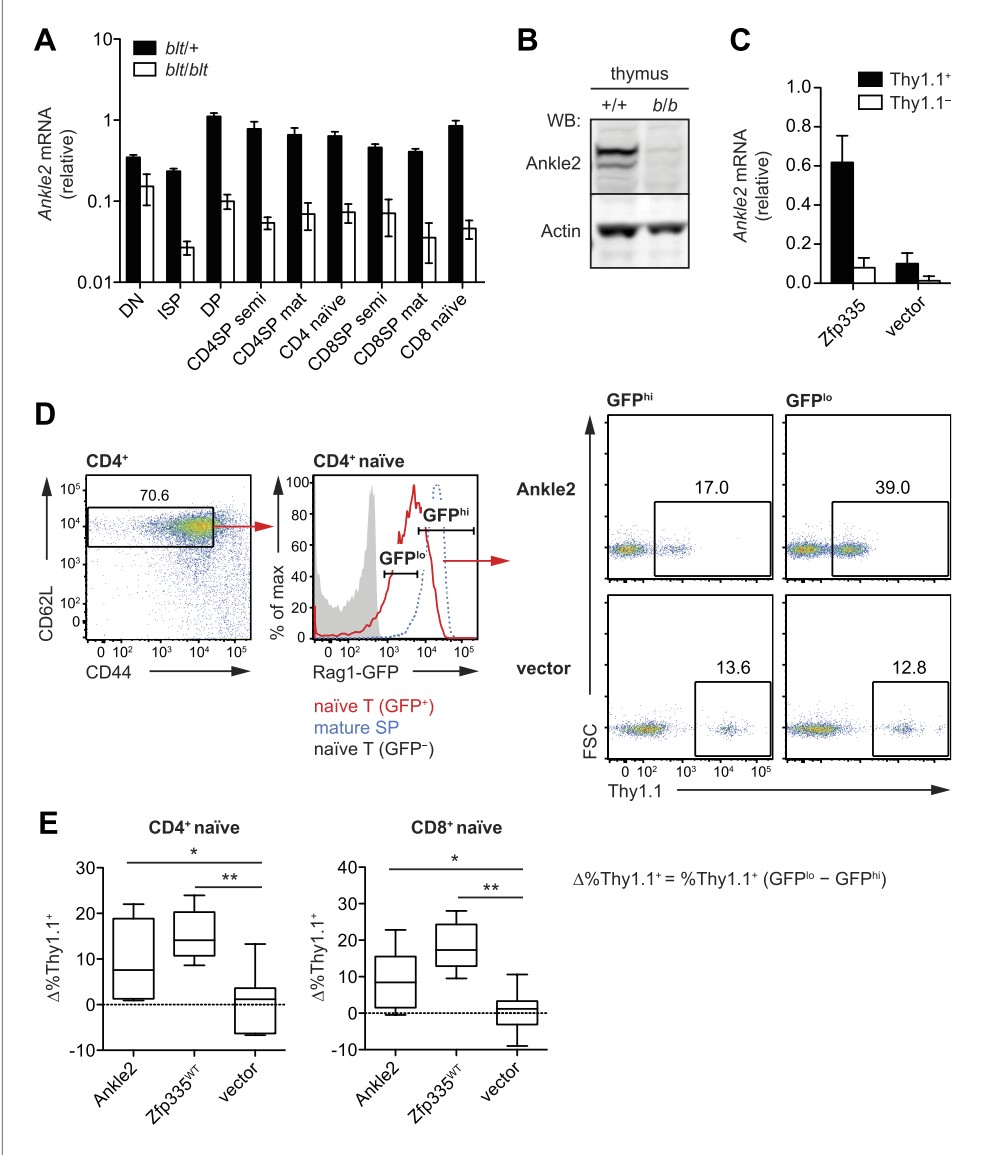

**Figure 8**. Ankle2 is a functional target gene of Zfp335 and its dysregulation by Zfp335R1092W contributes to the maturation defect in blt/blt T cells. (**A**) Quantitative RT-PCR analysis of *Ankle2* transcript levels in indicated thymocyte and naïve T cell populations sorted from *blt/+* and *blt/blt* mice (mean ± s.d., $n$ = 3–4). (**B**) Western blot for Ankle2 protein in wild-type (+/+) and mutant (*b/b*) thymocytes, with actin as loading control. (**C**) Rag1-GFP *blt/blt* bone marrow was retrovirally transduced with either WT Zfp335 or control vector and used to reconstitute irradiated hosts. CD4+ RTEs from these chimeras were sorted into transduced (Thy1.1+) and non-transduced (Thy1.1−) populations and analyzed for *Ankle2* expression by RT-qPCR (mean ± s.d., $n$ = 3). (**D**) Gating strategy for spleen CD4+ naïve T cells, subdivided into GFPhi (less mature) and GFPlo (more mature) populations. Red line, Rag1-GFP+ naïve T cells; blue dashed line, Rag1-GFP+ mature SP thymocytes; grey fill, Rag1-GFP− CD45.1+ host naïve T cells (background control). GFPlo and GFPhi T cells were then gated on Thy1.1 reporter expression as indicated. Flow cytometry plots shown are for chimeras reconstituted with Ankle2- or empty vector-transduced Rag1-GFP *blt/blt* bone marrow. (**E**) Change in the percentage of cells transduced with Ankle2 ($n$ = 6), WT Zfp335 ($n$ = 5) or control vector ($n$ = 10) during naïve T cell maturation (Δ%Thy1.1+ = %Thy1.1+ GFPlo − %Thy1.1+ GFPhi). A higher Δ%Thy1.1+ indicates enrichment of the reporter+ cells in the more mature GFPlo population compared to the less mature GFPhi population. Data are represented as Tukey box plots; *$p$ < 0.05, **$p$ < 0.01 (one-tailed Mann–Whitney test). Chimeras were analyzed 8 weeks post-reconstitution.

The following figure supplement is available for figure 8:

**Figure supplement 1**. Ectopic expression of other Zfp335 target genes is not sufficient to reverse the T cell maturation defect.

to minimize the harmful impact of reactive oxygen species (ROS) as newly mature cells transition from the thymus to the periphery (*Pua et al., 2009*). Although we observed genes involved in oxidative phosphorylation enriched amongst Zfp335 targets, our gene expression data did not reveal a clear autophagy signature, and initial experiments found no significant differences in mitochondrial content or ROS levels in *blt/blt* mature SP thymocytes (data not shown). However, we cannot rule out the possibility that other autophagy-dependent processes may be affected by Zfp335 deficiency.

From our microarray analyses, we found that most gene expression changes in *blt/blt* T lymphocytes were subtle. In our mature CD4SP thymocyte data set, of the 1504 genes passing a p-value threshold of <0.05, only 16 genes, or 1% of the total had a twofold or greater change in expression, while the vast majority (93%) showed a less than 1.5-fold difference between *blt/blt* and WT. This may not be a surprising result, given that we were only able to detect strongly diminished $Zfp335^{R1092W}$ occupancy at a limited subset of target sites. However, due to the highly conservative criteria used and other technical limitations discussed earlier, this is most likely an underestimate of the true degree of differential binding, which may affect a broader range of target genes and account for many of the more subtle changes in gene expression. Consistent with this notion, we observed that 33 direct Zfp335 targets, representing ~20% of the total, were differentially expressed in mature CD4SP thymocytes (p < 0.05), and of these genes, only four (*Ankle2*, *Cnpy2*, *Nme6*, *Tbck*) met our differential binding criteria. Furthermore, several known or putative transcriptional regulators are among genes directly targeted by Zfp335 (*Supplementary file 4*) or differentially expressed in *blt/blt* thymocytes and RTEs (*Supplementary file 1*), suggesting that Zfp335 operates within the context of a broader gene regulatory network. It is likely that the T cell maturation defect arises from the cumulative effects of relatively small gene expression changes within the Zfp335 regulatory network, possibly resulting in mild perturbations of multiple pathways whose functional consequences may not be individually significant but in combination may negatively impact overall T cell fitness.

It is interesting to note that survival of mature naïve T cells appears to be intact, despite RT-qPCR evidence indicating that a number of Zfp335 target genes found to be downregulated in *blt/blt* mature SP thymocytes and RTEs (*Ankle2*, *Cnpy2*, *Sep15*, and *Wdr47*) are similarly downregulated in the total naïve T cell population (*Figure 8A* and data not shown). If we extend this observation to the genome-wide level to assume that $Zfp335^{R1092W}$-induced transcriptional dysregulation in mature naïve T cells approximates the situation in RTEs, one way to explain why mature T cells are less affected could be that the deregulated genes and their associated pathways are most critical during late-thymic to early post-thymic maturation, after which they become dispensable. In support of this concept, it has been reported that IKK2 is required transiently in RTEs but not in mature T cells for normal IL-7Rα upregulation and homeostasis (*Silva et al., 2014*), although we have no evidence that this pathway is affected in *blt/blt* T lymphocytes. Alternatively, stochastic variations in gene expression or upregulation of compensatory mechanisms may have resulted in comparatively 'fitter' cells being preferentially selected into the mature naïve T cell pool.

Zfp335 is broadly expressed in hematopoietic cells (*Heng et al., 2008*) and in a variety of non-lymphoid tissues such as brain, kidney, heart, and lung (*Wu et al., 2009*). Given its widespread expression and the early embryonic lethality resulting from its complete ablation (*Yang et al., 2012*), it is highly likely that Zfp335 serves wider developmental roles. The poor contribution of Zfp335-overexpressing cells to hematopoietic lineages in BM chimeric mice also hints at a wider role. It may therefore seem remarkable that a mutation in such a ubiquitously expressed gene could produce a defect that appears to be selective for this particular stage of T cell development. It is possible that the loss of Zfp335 binding to the target sites described in our study may be unique to T lymphocytes, resulting in T cell-specific dysregulation; however, this hypothesis is not favored by preliminary evidence showing that target genes such as *Ankle2* were also downregulated in B cells (data not shown). As discussed earlier, our data that only a small number of target genes are significantly deregulated suggests that other aspects of the Zfp335-dependent transcriptional program may remain sufficiently intact to allow most developmental processes, except for T cell maturation, to proceed relatively normally. For future studies, it will be important to test whether a null allele has additional consequences for immune regulation beyond the selective T cell maturation effects revealed by the *bloto* mutation.

Genome-wide, we estimate that there are more than 2000 promoter regions containing Zfp335 motif sites. Our study detected approximately 150 ChIP-seq peaks in thymocytes, consistent with the generally accepted principle that additional molecular requirements beyond DNA sequence preference determine the stability of transcription factor binding to a given site. At this early stage, it is not

known what these requirements are for Zfp335, but they are likely to involve local chromatin context and interactions with co-binding partners which remain to be defined. It is also unclear whether Zfp335 displays cell type-specific binding patterns, which may allow Zfp335 to fulfill various developmental roles by regulating different gene expression programs. Finally, although it has been proposed that Zfp335 activates target genes by recruiting H3K4 methyltransferases (*Yang et al., 2012*), it is not clear if this is the sole mechanism, or that Zfp335 functions exclusively as a transcriptional activator. Consistent with the view that Zfp335 drives transcriptional activation, we observed that most target genes with deregulated expression in *blt/blt* thymocytes tended to be downregulated. However, we have noted exceptions in which some target genes—including Zfp335 itself—were modestly upregulated, suggesting the possibility that Zfp335 may also act as a repressor. Further investigations will be needed to identify the interaction partners with which Zfp335 cooperates to control gene expression.

To conclude, our findings regarding Zfp335$^{R1092W}$ add to other recent work (*Hsu et al., 2011*; *Landry et al., 2011*; *Silva et al., 2014*) to highlight unique gene expression requirements in late stage thymocytes and recent thymic emigrants for the formation of a normal sized naïve T cell compartment. In this regard, the function of Zfp335 is similar to its action in the brain where it is needed for the formation of a normal sized forebrain structure (*Yang et al., 2012*). In future work, it will be important to build from these findings to understand how this broad set of gene expression changes are integrated to promote formation of a normal sized compartment of cells. It will also be important to understand whether Zfp335 acts constitutively during late stages of T cell development or whether its function is regulated by external inputs and thus serves as a checkpoint that can influence the size and properties of the naïve T cell compartment.

## Materials and methods

### Mice

The Zfp335$^{bloto}$ strain was established through ethylnitrosourea (ENU)-mediated mutagenesis of C57BL/6 (B6) mice at the Australian National University using methods previously described (*Randall et al., 2009*). Putative mutants were identified as having blood CD4$^+$ and CD8$^+$ T cell frequencies more than one standard deviation below the mean. CD45.1$^+$ congenic mice were from the National Cancer Institute (01B96; B6-LY5.2/Cr). CD45.1$^+$CD45.2$^+$ mice were generated by crossing B6 and Boy/J (Jackson Laboratory, 002014; B6.SJL-*Ptprc$^a$Pepc$^b$*/BoyJ) mice. OTII TCR transgenic mice (Tg[TcraTcrb]426-6Cbn) were from an internal colony. Rag1-GFP transgenic mice were provided by N Sakaguchi (Kumamoto University, Kumamoto, Japan) (*Kuwata et al., 1999*). Lck-BCL2 transgenic mice were generated by S Korsmeyer (Dana-Farber Cancer Institute, Boston, MA) (*Sentman et al., 1991*) and provided by A Winoto (University of California Berkeley, Berkeley, CA). RIP-mOVA transgenic mice (Tg[Ins2-OVA]59Wehi) were provided by S Sanjabi (University of California San Francisco). Mice were housed in specific pathogen-free conditions, and all experiments were done according to the Institutional Animal Care and Use Committee guidelines of the University of California San Francisco.

### Genetic mapping and sequencing of the *bloto* mutation

Affected *bloto* mice were crossed onto the CBA/J background to generate heterozygous F1 mice. F1 mice were intercrossed to yield F2 progeny homozygous for the *bloto* mutation and carrying a mix of C57BL/6 and CBA/J single nucleotide polymorphisms (SNPs). SNP mapping using an Amplifluor assay (EMD Millipore, Billerica, MA) with Platinum Taq (Life Technologies, Carlsbad, CA) was carried out on genomic DNA isolated from affected and unaffected mice. Exome enrichment was performed using the SeqCap EZ Mouse Exome kit (Roche Nimblegen, Basel, Switzerland), followed by 75 bp paired-end sequencing on the Illumina Genome Analyzer IIx platform (Illumina, San Diego, CA). Computational analysis to detect novel single-nucleotide variants was done as previously described (*Andrews et al., 2012*). The affected exon was PCR-amplified from genomic DNA and Sanger sequencing (TACGen, Richmond, CA) was carried out to confirm the mutation.

### Genotyping

Zfp335$^{bloto}$ mice were genotyped by allele-specific PCR using the following primers: WT-F: 5′-AGAACAAGAAGGATCTGAGGC-3′; bloto-F: 5′-AAGAACAAGAAGGATCTGAGGT-3′; common-R: 5′-GGCTCGGGCTGTAGAAGT-3′. WT and *bloto* allele-specific primers were run in separate reactions with GoTaq Hot Start polymerase (Promega, Madison, WI).

## Constructs

Full-length Zfp335 was cloned from cDNA into an MSCV retroviral vector containing a Thy1.1 (CD90.1) reporter downstream of an internal ribosome entry site (IRES). The *bloto* mutation (c.3274C > T) was introduced by site-directed mutagenesis. For use in transfections, WT and *bloto* Zfp335 inserts were subcloned into a pcDNA3.1 vector (Life Technologies) with a FLAG epitope tag at the N-terminus. Ankle2, Cnpy2, Nme6, Sep15, Tbck, and Wdr47 were PCR amplified from mouse cDNA and cloned into the MSCV-IRES-Thy1.1 vector. All constructs were verified by sequencing. Reference sequences used in this study are as follows—protein: Zfp335 (NP_950192.2); mRNA: Zfp335 (NM_199027.2), Ankle2 (NM_001253814.1), Cnpy2 (NM_019953.1), Nme6 (NM_018757.1), Sep15 (NM_053102.2), Tbck (NM_001163455.1), Wdr47 (NM_181400.3).

## Flow cytometry

Cells were isolated from thymus, spleen, and lymph nodes by mechanical disaggregation through a 40-μm nylon sieve and stained as described (*Schmidt et al., 2013*). Antibodies were as follows: anti-CD4 (GK1.5, RM4-5), CD8 (53–6.7) (Biolegend, San Diego, CA; Tonbo Biosciences, San Diego, CA); CD62L (MEL-14), CD44 (IM7), CD69 (H1.2F3), CD24 (M1/69), CD45.1 (A20), CD45.2 (104), TCRβ (H57-597), CD19 (6D5), TCRγδ (GL3), Thy1.1 (OX-7), CD25 (PC61) (Biolegend); NK1.1 (PK136), Vα2 (B20.1), CD5 (53-7.3) (BD Biosciences, San Jose, CA); IL-7Rα (A7R34), Foxp3 (FJK-16 s), BrdU (BU20A) (eBioscience, San Diego, CA); mCD1d/PBS-57 tetramer (NIH Tetramer Core Facility). 4′,6-diamidino-2-phenylindole (DAPI) was used for dead cell exclusion. For all intracellular staining, cells were stained for surface antigens before fixation. Foxp3 was stained using the Foxp3 Staining Buffer Set (eBioscience). BrdU staining was performed as per manufacturer's guidelines (BD Biosciences). For cell cycle analysis by DNA content, cells were fixed with BD Cytofix/Cytoperm Buffer and stained with 5 μM DAPI in Perm/Wash buffer (BD Biosciences). Annexin V staining was performed using the Annexin V-PE Apoptosis Detection kit (BD Biosciences) according to manufacturer's instructions. Samples were acquired on an LSRII cytometer (BD Biosciences) and analyzed with FlowJo software (TreeStar, Ashland, OR).

## Cell sorting

For microarray analysis, hematopoietic chimeras were generated with a mix of CD45.1$^+$CD45.2$^+$ Rag1-GFP WT and CD45.2$^+$ Rag1-GFP *blt/blt* bone marrow. Thymocytes were isolated in MACS buffer (PBS +2% FBS, 2 mM EDTA), incubated with anti-CD8 microbeads (Miltenyi Biotec, Bergisch Gladbach, Germany), and MACS-depleted to enrich for CD4SP thymocytes, after which cells were stained with anti-CD45.1, CD4, CD8, CD69, and CD62L. Mature CD4SP thymocytes (GFP$^{hi}$CD4$^+$CD8$^-$CD62L$^{hi}$CD69$^{lo}$) were sorted into WT (CD45.1$^+$) and *blt/blt* (CD45.1$^-$) subsets. For RTE sorting, spleen, and lymph node cells were pooled and erythrocytes were lysed in a solution of Tris-buffered NH$_4$Cl. Cells were labeled with anti-CD45.1, CD4, CD8, CD62L, and CD44. RTEs were sorted as CD4$^+$CD62L$^{hi}$CD44$^{lo}$ Rag1-GFP$^{hi}$ CD45.1$^+$ (WT), and CD45.1$^-$ (*blt/blt*) subsets. Dead cells were excluded with DAPI. Samples were sorted on a FACSAria with ≥98% purity.

## Immunofluorescence

Mature CD4SP thymocytes were sorted and allowed to adhere to poly-L-lysine-coated glass slides (P0425; Sigma–Aldrich, St. Louis, MO). Cells were then fixed with 4% PFA in PBS and permeabilized with 0.1% Triton X-100 for 10 min on ice, followed by blocking (5% normal goat serum, 2% BSA in PBS) and staining (2% goat serum, 0.1% BSA, 0.1% Tween-20 in PBS). For Zfp335 detection, slides were incubated at RT with primary antibody (A300-797A, A300-798A; Bethyl Laboratories, Montgomery, TX) for at least 3 hr, followed by biotin goat anti-rabbit (BD Biosciences) and lastly SA-Cy3 (Jackson Immunoresearch, West Grove, PA). Slides were counterstained with 1 μM DAPI and mounted with Fluoromount-G (Southern Biotech, Birmingham, AL). Confocal imaging was performed on a Leica SP5 inverted microscope with a 63× oil immersion objective. Images were processed with Leica LAS software.

## Bone marrow chimeras

Mixed bone marrow chimeras were generated by intravenously transferring $3 \times 10^6$ to $5 \times 10^6$ cells from the following mixes into lethally irradiated ($2 \times 450$ rads, 3 hr apart) CD45.1$^+$ congenic mice: *blt/blt* CD45.2$^+$: WT CD45.1$^+$CD45.2$^+$; *blt/+* CD45.2$^+$: WT CD45.1$^+$CD45.2$^+$, at a 50:50 ratio. For OTII/ RIP-mOVA experiments, bone marrow from OTII *blt/+* or OTII *blt/blt* mice was incubated with

biotin-anti-CD3, CD4, and CD8 followed by anti-biotin microbeads (Miltenyi Biotec) and T cells were depleted by MACS prior to transfer into lethally irradiated WT B6 or RIP-mOVA recipients. Retroviral transduction of bone marrow from *blt/blt* or Rag1-GFP *blt/blt* mice was performed as described (*Green et al., 2011*) using the Platinum-E packaging cell line. Chimeric mice were analyzed at least 8 weeks after reconstitution.

## Adoptive transfers

For peripheral T cell transfers, spleen and lymph node cells were harvested from WT (control) or *blt/blt* mice and RBC-lysed. For thymocyte transfers, bulk thymocytes were prepared from *blt/+* (control) or *blt/blt* mice. To distinguish between the two populations, either control or *blt/blt* cells were labeled with 1 μM CFSE (Life Technologies) for 10 min at 37°C before they were mixed and intravenously injected into lymphoreplete Boy/J (CD45.1+) mice. Recipients were analyzed at days 1 and 7 post-transfer and the percentage of *blt/blt* naïve T cells in the CD45.2+ donor population was determined. For Rag1-GFP T cell transfers, spleen and lymph node cells were harvested from Rag1-GFP *blt/+* or Rag1-GFP *blt/blt* mice and RBC-lysed. Non-transgenic CD45.2+ *blt/+* cells were used as a mixing control. Either control or Rag1-GFP cells were labeled with 10 μM CellTrace Violet (Life Technologies) for 20 min at 37°C before mixing and intravenous transfer into Boy/J recipients. At days 1, 3, and 5 post-transfer, Rag1-GFP *blt/+* or Rag1-GFP *blt/blt* naïve T cells were assessed as a proportion of the total CD45.2+ donor population.

## BrdU and FTY720 treatment

For BrdU experiments, mice received two intraperitoneal (i.p.) injections of 1 mg BrdU spaced 2 hr apart and were euthanized 4 hr after the first dose, or were fed 1 mg/ml BrdU in drinking water for longer-term labeling. To block thymic egress and induce mature SP thymocyte accumulation, mice were injected i.p. twice with FTY720 (Cayman Chemical, Ann Arbor, MI) dissolved in saline at a dose of 1 mg/kg body weight on days 0 and 2, and sacrificed on day 4.

## T cell stimulation and culture

FACS-purified CD4+ naïve T cells were labeled with 5 μM CFSE for 10 min at 37°C, quenched with fetal bovine serum (FBS) and washed in 10% FBS. Labeled cells were stimulated with plate-bound anti-mouse CD3 (clone 2C11; 1 μg/ml) and anti-mouse CD28 (clone 37.51; 1 μg/ml) for 3 days. For in vitro cell viability assays, semi-mature and mature CD4SP thymocytes were sorted from chimeras reconstituted with a mix of either *blt/+* and WT or *blt/blt* and WT BM. Sorted cells were cultured in a 96-well plate at a density of 6–10 × 10⁴ cells per well. All cells were cultured at 37°C in 5% $CO_2$, using RPMI media supplemented with 10% FBS, L-glutamine, β-mercaptoethanol, penicillin, and streptomycin.

## Western blotting

Thymocytes were lysed in RIPA buffer containing protease inhibitor cocktail (EMD Millipore). Samples were resolved on NuPAGE Bis-Tris gels (Life Technologies) and transferred to Immobilon-FL membranes (EMD Millipore). Primary antibodies used: anti-Zfp335 (A300-797A, A300-798A; Bethyl Laboratories), anti-actin (A2066, Sigma–Aldrich), and anti-FLAG M2 (F1804, Sigma–Aldrich). Ankle2 was detected using rabbit antiserum raised against human ANKLE2, provided by I Mattaj (EMBL, Heidelberg, Germany) (*Asencio et al., 2012*). Secondary antibodies used: goat anti-mouse IRDye 800CW, donkey anti-rabbit IRDye 700DX (Rockland Immunochemicals, Gilbertsville, PA). Blots were scanned using the Odyssey Infrared Imaging System (LI-COR Biosciences, Lincoln, NE).

## Electrophoretic mobility shift assay (EMSA)

HEK293T cells were transfected with FLAG-tagged WT or *bloto* Zfp335 (cloned as described) and nuclear extracts prepared using a modified Dignam protocol. Relative protein amounts were determined by Western blot using anti-Zfp335 (A300-797A). Gel mobility shift assays were performed as described (*Lo et al., 1991*). Briefly, nuclear extracts were mixed with 10 mM HEPES pH 7.9, 0.1 mM EDTA, 10% glycerol, 0.5 mM DTT, 0.1 mM PMSF, 0.08 mg/ml BSA, 0.04 mg/ml poly dI/dC, 0.4 mM $ZnCl_2$, and 1' mM biotin-labeled probe and incubated at room temperature for 20 min. For experiments using competitor oligos, 20- (*Figure 7F*) or 50-fold (*Figure 7G*) molar excess of unlabeled probe was pre-incubated with nuclear extracts on ice for 30 min before adding biotinylated probe. Samples were loaded on a 4% polyacrylamide gel with 3.5% glycerol in 0.25X TBE and run at constant voltage in 0.25X TBE running buffer, then transferred to Biodyne B nylon membranes (Pall Corporation, Port Washington, NY) in 0.5X TBE at 380 mA for 1 hr. Signal was detected with LightShift Chemiluminescent

Nucleic Acid Detection Module kit (Thermo Scientific, Waltham, MA). All probes were derived from synthetic double-stranded 5'-biotinylated oligonucleotides, listed in *Supplementary file 5* (Integrated DNA Technologies, San Jose, CA).

## Quantitative RT-PCR and microarray analysis

Total RNA was isolated using the RNeasy Micro Kit (Qiagen, Venlo, The Netherlands); cDNA was synthesized using MMLV reverse transcriptase and random primers (Life Technologies) according to manufacturer's instructions. Real-time PCR was carried out using a StepOnePlus real-time PCR system (Applied Biosystems, Foster City, CA) with SYBR Green PCR Master Mix (Applied Biosystems) and the appropriate primer pairs (*Supplementary file 6*; Integrated DNA Technologies). Relative mRNA abundance of target genes was determined by subtracting the threshold cycle for the internal reference (*Hprt1*) from that of the target. Primer pairs were tested for linear amplification over two orders of magnitude.

For microarray analysis, cDNA was prepared using the Ovation Pico WTA System V2 and labeled using the Encore Biotin Module (NuGEN, San Carlos, CA). Labeled cDNA libraries were hybridized to Affymetrix Mouse 1.0 ST arrays (Affymetrix, Santa Clara, CA) and scanned with the Affymetrix GeneChip Scanner 3000 7G System. Raw data were normalized by RMA and probesets mapped to unique Entrez Gene IDs using a custom Brainarray CDF. The limma R package was used for analyzing differential gene expression. GENE-E software (Broad Institute) was used for heat map generation and hierarchical clustering.

## ChIP-qPCR

Total thymocytes ($20 \times 10^6$) or CD4SP thymocytes ($4$–$10 \times 10^6$) were harvested from WT and *blt/blt* mice, cross-linked in 1% formaldehyde for 10 min at room temperature, quenched with 125 mM glycine and washed twice in ice-cold PBS. Cells were lysed in buffer containing 20 mM Tris–HCl (pH 8.0), 85 mM KCl, 0.5% Nonidet-P40, followed by nuclear lysis buffer (50 mM Tris–HCl pH 8.0, 10 mM EDTA, 1% SDS). Chromatin was sonicated with a Bioruptor (Diagenode, Liège, Belgium), cleared by centrifugation and diluted in buffer containing 20 mM Tris–HCl (pH 8.0), 1.1 mM EDTA, 140 mM NaCl, 0.01% SDS, and 1.1% Triton X-100. Diluted chromatin was incubated overnight at 4°C with antibody bound to Protein A Dynabeads (Life Technologies). All buffers up to this point were supplemented with protease inhibitors. Beads were washed in low salt buffer (10 mM Tris–HCl pH 8.0, 150 mM NaCl, 1 mM EDTA, 1% Triton X-100), high salt buffer (20 mM Tris–HCl pH 8.0, 500 mM NaCl, 2 mM EDTA, 1% Triton X-100, 0.1% SDS), LiCl buffer (10 mM Tris–HCl pH 8.0, 250 mM LiCl, 1 mM EDTA, 0.5% sodium deoxycholate, 0.5% Nonidet-P40), and TE buffer. Protein/DNA complexes were eluted (50 mM Tris–HCl pH 8.0, 10 mM EDTA, 1% SDS) at 65°C for 30 min with shaking on a Thermomixer (Eppendorf). Eluted complexes were reverse-crosslinked overnight at 65°C and then treated with RNase A (Sigma-Aldrich) and proteinase K (Life Technologies). ChIP DNA was purified using the QIAquick PCR purification kit (Qiagen) and quantitative PCR was performed using SYBR Green PCR Master Mix (Applied Biosystems) and primer pairs listed in *Supplementary file 7*. For Zfp335 ChIP, a polyclonal antibody specific for a C-terminal epitope (A300-798A; Bethyl Laboratories) was used.

## ChIP-seq

ChIP was performed with total thymocytes ($60 \times 10^6$ cells) as described above. Two antibodies were used: one raised against a C-terminal epitope (A300-798A; Bethyl Laboratories) and the other against an N-terminal epitope (A300-797A; Bethyl Laboratories) of ZNF335. 7–10 ng ChIP DNA and 10 ng input DNA were used for library preparation, performed according to Illumina's TruSeq protocol with some modifications. DNA clean-up, removal of adapter dimers, and size selection were done using Agencourt AMPure XP beads (Beckman Coulter, Brea, CA). Libraries were checked for quality using the High Sensitivity DNA Bioanalyzer kit (Agilent Technologies, Santa Clara, CA), quantified with the Qubit dsDNA HS Assay kit (Life Technologies), and sequenced as 50 bp single-end reads on the Illumina HiSeq 2000 platform.

## ChIP-seq data analysis

Illumina adapter sequences were removed using the cutadapt tool. Trimmed reads were aligned to the mm9 reference genome using bwa, allowing for a maximum of two mismatches. Reads aligned with a MAPQ score of less than 20 were filtered out using samtools. Basic peak calling was performed with MACS2 (parameters: -g mm --bw = 300 –q 0.05). Peaks were annotated with their nearest RefSeq TSS using HOMER. Genomic feature annotation summary statistics were generated using the Galaxy/Cistrome

CEAS module (version 1.0.0), and the full set of target genes (*n* = 177) defined as having a TSS within 1 kb of high-confidence peaks were identified using the BETA-minus module (version 1.0.0). To call differential binding events, samtools rmdup was first used to remove duplicate reads, after which regions of differential enrichment were identified using the MACS2 callpeak and bdgdiff modules. Normalized pileup tracks were generated from nonredundant reads using MACS2 callpeak--SPMR (fragment pileup per million reads) and converted to the bigWig format for visualization on the UCSC Genome Browser. The C-terminal antibody (A300-798A) was found to give better enrichment so we based our analyses of general Zfp335 binding properties on the ChIP-C WT data set, unless otherwise stated. H3K4me3, H3K27ac, and H3K27me3 ChIP-seq data for mouse thymus were downloaded from ENCODE/LICR (*ENCODE Project Consortium, 2012*), and aggregate density profiles were computed using bwtool.

## Gene ontology and pathway analysis

Gene ontology enrichment analysis of Zfp335 binding regions was performed using the Genomic Regions Enrichment of Annotations Tool (GREAT). Each ChIP-seq peak was associated with the two nearest genes within 10 kb.

## Motif analysis

400 bp sequences centered on peak summits were extracted from the 119 top-scoring WT Zfp335 peak regions, repeat-masked and used for de novo motif discovery with MEME (parameters: min. width = 6, max. width = 30, zero or one instances of a given motif per sequence). The top-scoring motif obtained with MEME was replicated using HOMER's motif finding function. HOMER was used to detect Zfp335 motif occurrences and locations within defined genomic regions. HOMER was used to generate a histogram of motif density (bin size = 50 bp) for a region from −500 bp to +500 bp of Zfp335 ChIP-seq peaks. DNase I-seq signal and phyloP conservation scores for motifs occurring within Zfp335 peaks or ±2 kb of RefSeq TSS were aggregated using bwtool. The DNase I digital genomic footprinting (DGF) signal track for mouse thymus was obtained from ENCODE/UW (*ENCODE Project Consortium, 2012*); sequence conservation tracks were downloaded from UCSC Genome Browser (*Raney et al., 2014*).

## Bioinformatics

Software tools used:

- R 3.0.2
- affy R package (*Gautier et al., 2004*)
- limma R package (*Smyth, 2004*)
- Brainarray custom CDF (mogene10st_Mm_ENTREZG_17.1.0) (*Dai et al., 2005*)
- cutadapt 1.4.1 (*Martin, 2011*)
- bwa 0.7.7 (*Li and Durbin, 2009*)
- samtools 0.1.18 (*Li et al., 2009*)
- bedtools 2.17.0 (*Quinlan and Hall, 2010*)
- bwtool (*Pohl and Beato, 2014*)
- MACS 2.0.10 (*Zhang et al., 2008*)
- MEME 4.9.1 (*Bailey et al., 2009*)
- HOMER 4.5 (*Heinz et al., 2010*)
- GREAT 2.0.2 (*McLean et al., 2010*)
- GSEA 2.0.14 (*Subramanian et al., 2005*)
- Galaxy/Cistrome (*Liu et al., 2011*)
- UCSC Genome Browser (http://genome.ucsc.edu) (*Kent et al., 2002*, *2010*; *Rosenbloom et al., 2012*)

## Protein structural modeling

A homology model was generated by SWISS-MODEL utilizing using the known structure of a designed DNA-binding zinc finger protein (Protein Data Bank ID: 1MEYC) as template. Figures were created using MacPyMOL (version 1.5.0.5).

## Statistical analysis

Data were analyzed with Prism 5 (GraphPad Software). The two-tailed non-parametric Mann–Whitney test was used for comparison of two unpaired groups for all data sets unless otherwise indicated.

## Data availability

Microarray and ChIP-seq data sets generated in this study were deposited to NCBI's Gene Expression Omnibus under SuperSeries GSE58293.

## Acknowledgements

We thank A Winoto (University of California Berkeley) and S Sanjabi (University of California San Francisco) for mice, J Landry (Virginia Commonwealth University) for sharing *Bptf*[−/−] expression data, H Schjerven, G Yeh, and S Smale (University of California Los Angeles) for EMSA protocols and advice, P Hartley and E Dang for ChIP-seq protocols, and M Thomson and A Marson (University of California San Francisco) for bioinformatics discussion. We acknowledge the ENCODE Consortium for histone ChIP-seq and DNase I-seq data sets, and the NIH Tetramer Core Facility (contract HHSN272201300006C) for provision of the mCD1d/PBS-57 tetramer. Special thanks to M Ansel for critical reading of the manuscript, Y Xu and J An for technical assistance, and members of the Cyster lab for helpful discussion.

## Additional information

### Funding

| Funder | Grant reference number | Author |
| --- | --- | --- |
| National Institutes of Health | R01 AI74847 | Jason G Cyster |
| Howard Hughes Medical Institute | | Jason G Cyster |
| Agency for Science, Technology and Research (A*STAR) | National Science Scholarship (PhD) | Brenda Y Han, Chuan-Sheng Foo |
| National Institutes of Health | R01 AI52127 | Chris C Goodnow |

The funders had no role in study design, data collection and interpretation, or the decision to submit the work for publication.

### Author contributions

BYH, Designed and performed experiments, Analyzed and interpreted data, Did bioinformatics analysis, Wrote the manuscript; SW, Performed experiments and analyzed data; C-SF, Performed bioinformatics analysis; RMH, Provided bioinformatics support; CNJ, Carried out ENU mutagenesis screening and genetic mapping, Conception and design; SRW, Carried out ENU mutagenesis screening; BW, Coordinated exome sequencing, Acquisition of data, Analysis and interpretation of data; CCG, Supervised ENU mutagenesis screening and exome sequencing, Conception and design; JGC, Planned the project, Designed the experiments, Interpreted data, Revised the manuscript

### Ethics

Animal experimentation: All animal handling and experimentation were performed according to approved Institutional Animal Care and Use Committee (IACUC) protocols of the University of California San Francisco (UCSF; approval number AN087331-03) and of the Australian Phenomics Facility and the Australian National University (ANU; approval number A201¼6).

## Additional files

### Supplementary files

• Supplementary file 1. Hierarchical clustering analysis of 108 genes differentially expressed (p < 0.05, fold-change > 1.2) in *blt/blt* vs. WT mature CD4SP thymocytes and CD4[+] RTEs.

• Supplementary file 2. Summary statistics for ChIP-seq read alignment, mapping and peak calling.

• Supplementary file 3. ChIP-seq peaks for all WT and bloto thymocyte datasets, annotated with the nearest gene TSS.

- Supplementary file 4. Functional annotation of Zfp335 target genes, with major enriched categories represented.
- Supplementary file 5. Sequences of oligonucleotides used for EMSA.
- Supplementary file 6. Sequences of primers used for RT-qPCR.
- Supplementary file 7. Sequences of primers used for ChIP-qPCR.

### Major datasets

The following datasets were generated:

| Author(s) | Year | Dataset title | Dataset ID and/or URL | Database, license, and accessibility information |
|---|---|---|---|---|
| Han BY, Wu S, Foo CS, Horton RM, Craig NJ, Watson SR, Whittle B, Goodnow CC, Cyster JG | 2014 | Gene expression profiling of mature CD4 SP thymocytes from a mouse strain with an ENU-induced mutation in Zfp335 | GSE58293; http://www.ncbi.nlm.nih.gov/geo/query/acc.cgi?acc=GSE58288 | Publicly available at NCBI Gene Expression Omnibus. |
| Han BY, Wu S, Foo CS, Horton RM, Craig NJ, Watson SR, Whittle B, Goodnow CC, Cyster JG | 2014 | Gene expression profiling of recent thymic emigrants (RTEs) from a mouse strain with an ENU-induced mutation in Zfp335 | GSE58293; http://www.ncbi.nlm.nih.gov/geo/query/acc.cgi?acc=GSE58289 | Publicly available at NCBI Gene Expression Omnibus. |
| Han BY, Wu S, Foo CS, Horton RM, Craig NJ, Watson SR, Whittle B, Goodnow CC, Cyster JG | 2014 | Zfp335 ChIP-seq in WT and blt/blt thymocytes | GSE58293; http://www.ncbi.nlm.nih.gov/geo/query/acc.cgi?acc=GSE58333 | Publicly available at NCBI Gene Expression Omnibus. |

The following previously published datasets were used:

| Author(s) | Year | Dataset title | Dataset ID and/or URL | Database, license, and accessibility information |
|---|---|---|---|---|
| ENCODE/LICR/Ren | 2012 | H3K27ac ChIP-seq on 8-week mouse thymus | wgEncodeEM002474 | Publicly available at ENCODE (https://www.encodeproject.org). |
| ENCODE/LICR/Ren | 2012 | H3K27me3 ChIP-seq on 8-week mouse thymus | wgEncodeEM002724 | Publicly available at ENCODE (https://www.encodeproject.org). |
| ENCODE/LICR/Ren | 2012 | H3K4me3 ChIP-seq on 8-week mouse thymus | wgEncodeEM002476 | Publicly available at ENCODE (https://www.encodeproject.org). |
| ENCODE/ UW/ Stamatoyannopoulos | 2012 | DNase-seq on 8 week mouse thymus | wgEncodeEM002912 | Publicly available at ENCODE (https://www.encodeproject.org). |

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
