## [Decision Letter]

Thank you for sending your work entitled “Zinc finger protein Zfp335 is required for
formation of the naïve T cell compartment” for consideration at *eLife.*
Your article has been favorably evaluated by Tadatsugu Taniguchi (Senior editor), a
member of our Board of Reviewing Editors, and 3 reviewers.

The referees’ comments, shown below, were both positive and constructive. After
consultation, Michel Nussenzweig, our Reviewing editor, has decided that the conditional
KO experiment suggested by referee #3 is not necessary.

*Reviewer 1*:

This work from Dr. Cyster’s group has identified Zfp335 as a novel regulator of naive T
cell maturation. The authors have elegantly combined forward genetics, immune assays and
transcriptional analysis, and the experiments were designed and performed well with
convincing results presented. This is certainly an interesting study, but I have the
following comments for the authors to address.

1) Although the mutant mice have reduced peripheral cells, the physiological
consequences on immune responses are unclear. The authors should test this in models of
T cell-dependent immune responses.

2) In Figure 3, the authors showed that thymic
but not peripheral naive T cells have defects in the maintenance upon adoptive transfer.
They then described defect in RTE as a likely mechanism, but have not directly tested
the survival or maintenance of RTE cells. This needs to be done.

3) In Figure 4, the authors measured expression
of Bcl2 family proteins to exclude a role of cell apoptosis. They should directly
measure cell apoptosis, e.g. by caspase staining.

4) In Figure 5, the authors used ChIP-Seq on
total thymocytes to identify Zfp335 targets. Since the most relevant cell type is mature
thymocytes, they need to validate the target genes by performing ChIP experiment using
mature thymocytes followed by qPCR of the targets.

*Reviewer 2*:

The manuscript by Han et al. describes the finding of a new Zinc finger protein Zfp335
in the development of mature thymocytes and peripheral T cells from the analysis of a
mouse strain from ENU-mutagenesis. The developmental defects of T cells are restricted
to the stages after DP thymocytes, particularly to mature SP, RTE, and peripheral T
cells.

The causative gene was found to be a hypomorphic mutation of Zfp335 from the experiment
by bone marrow reconstitution by retroviral transduction of the wild type and mutant
Zfp335. By very extensive analyses using various TCR-Tg mouse models and systems, the
authors suggested that the developmental defect could not be attributed to the impaired
selection, cell survival, and thymic egress. To elucidate the defect by identifying the
target genes of Zfp335, the authors found several possible target genes that Zfp335
directly binds. However, none of them could explain the developmental defect. One target
gene Ankle2 may partly restore mature T cells.

Overall the analysis of the mutant mouse and gene demonstrated a critical role of Zfp335
in development of T cells and identify the target genes, but failed to neither reveal
the mechanism of the developmental defect nor identify the function of Zfp335 gene.

1) Although Zfp335 restored mature T cell development by bone marrow reconstitution, it
is recommended to confirm by the knock-in of Zfp335 because the mice may still contain
other mutations and the heterogeneity of chimeric mice does not prove full restoration
of development.

2) Thymic selection using OTII-Tg mice showed significant defect of development of
mature TCR + T cells, the author might not simply neglect the effect on thymic
selection.

3) The data for the effect of Ankle2 in the reconstitution experiment was complicated
such as the analysis of Rag-GFPhigh and low population (for example why Thy1.1
expression levels were so different between Ankle2 vs. control). Simpler experiment and
expression should be taken to show significant restoration from developmental
arrest.

*Reviewer 3*:

In this manuscript, Han et al., describe the analysis of the T cell compartment in the
mouse mutant bloto, carrying a hypomorphic mutation in the zinc finger protein gene
Zfp335. The authors identify a missense mutation in Zfp335 that alters zinc finger 12
and affects the DNA-binding ability of the protein. Homozygous bloto mice show a defect
in the formation of the naive T cell compartment that cannot be attributed to altered
thymic selection, cell proliferation or survival. ChIP-seq and microarray analyses to
identify Zfp335-occupied and -regulated genes indicate that a very small number of genes
are differentially bound and regulated in mutant and wild type thymocytes.
Overexpression of one of these targets, Ankle2, in Rag1-GFP+ naive T cells showed a
partial rescue of the mutant phenotyope.

The extensive analysis of the T cell defect of Zfp335bloto mice and the molecular
examination of the defect in DNA binding and gene transcription are interesting and
extend previous work on the function of Zfp335 in neural stem cells (Cell 151, 1097,
2012). The data are convincing and well presented. In particular, the partial rescue of
the mutant phenotype by the overexpression of one of the identified Zfp335 targets
provides strong evidence for the functional role of this Zfp335-regulated gene. However,
the study would gain additional significance by the analysis of mice carrying a
conditional null allele of Zfp335. Such mice have been published, and ES cells that
harbor a conditional null allele of Zfp335 are available from a mutant mouse
repository.

1) The authors need to address and/or discuss the discrepancies between the previous and
current studies of Zfp335. For example, in Figure 4—figure supplement 3 the authors show that cell proliferation and survival are not affected by
the bloto mutation. However, the previous analysis of lymphoblastic cells of humans
carrying a hypomorphic H111R mutation shows an impaired growth of mutant cells.

2) In the ChIP-seq analysis, shown in Figures 5 and 6, the authors used the same anti-Zfp335 antibodies as the previous study.
However, the authors identified a different sequence motif than the previously reported.
Which parameters were used in the MEME analysis? What are the other significantly
enriched motifs? Did the analysis detect the motif described in the Cell paper? If the
motif width range is adjusted, is the other motif detected? I appreciate the inclusion
of an electrophoretic mobility shift assay, which provides strong evidence for the newly
identified sequence motif. The authors should also include the previously reported motif
in this analysis.

3) One of the strongest points of the paper is the partial rescue of the T cell defect
in bloto mice by the overexpression of Ankle2 (Figure 8). It would be of interest to examine the phenotype of an Ankle2
knock-down.

---

## [Author Response]

Reviewer 1:

*[…] This is certainly an interesting study, but I have the following comments
for the authors to address*.

*1) Although the mutant mice have reduced peripheral cells, the physiological
consequences on immune responses are unclear. The authors should test this in models
of T cell-dependent immune responses*.

We have tested the mice in different models of T cell-dependent immune responses, but
found no effect. In the first model, OTII *blt*/*blt* T
cells were able to expand and elicit an antigen-specific germinal center B cell response
from lysozyme specific Hy10 BCR transgenic B cells following duck egg lysozyme (DEL)-OVA
immunization, on par with OTII *blt*/+ controls. Similarly,
*blt*/*blt* mice showed no significant impairment in a
model of infection with PR8 influenza virus; they were able to mount a normal
NP_366-374_ tetramer-specific CD8^+^ T cell response despite having
lower numbers of peripheral T cells. These findings may be thought of as consistent with
our in vitro experiments showing that *blt*/*blt* naïve T
cells proliferate normally in response to TCR stimulation (Figure 4—figure supplement 3).

In summary, all the data we have accumulated up to this point suggest that the
*bloto* T cell defect is mainly developmental and does not profoundly
disrupt basic T cell function. Nevertheless, T lymphopenia negatively affects repertoire
size and is generally associated with an increased risk of infection. It is likely that
physiological consequences may be revealed when these mice are exposed to a broader
range of infectious agents than are typically present in a barrier facility, or if they
are infected with multiple pathogens simultaneously.

*2) In*
Figure 3*, the authors
showed that thymic but not peripheral naive T cells have defects in the maintenance
upon adoptive transfer. They then described defect in RTE as a likely mechanism, but
have not directly tested the survival or maintenance of RTE cells. This needs to be
done*.

We thank the reviewer for raising this important point. To address this concern, we have
performed additional adoptive transfer experiments of peripheral T cells using the
Rag1-GFP reporter system, and found that *blt*/*blt*
Rag1-GFP^+^ T cells underwent a steeper decline over time compared to
control *blt*/+ Rag1-GFP^+^ T cells. These results (new Figure 3) complement our analysis of thymic and
peripheral T cells in the original Figure 3 (now
moved to Figure 3—figure supplement 1) and
provide direct evidence for a defect in RTE maintenance. These data are discussed in the
revised text.

*3) In*
Figure 4*, the authors
measured expression of Bcl2 family proteins to exclude a role of cell apoptosis. They
should directly measure cell apoptosis, e.g. by caspase staining*.

We initially attempted annexin V and active caspase 3 staining in freshly isolated
thymocytes and naïve T cells ex vivo, but our data did not reveal clear differences and
were not included in our manuscript. It should be noted that the frequency of apoptotic
cells detected was extremely low, likely because dying cells are rapidly and efficiently
cleared in vivo, making it difficult to assess in vivo T cell death using these methods.
We now make this point in the revised text.

In response to the reviewer’s comment, we have analyzed the survival of sort-purified
mature CD4SP thymocytes following in vitro culture.
*blt*/*blt* cells showed an increased rate of cell
death (as determined by annexin V and DAPI staining) over time relative to co-cultured
WT controls, whereas the same effect was not observed in *blt*/+ mature
SP thymocytes (new Figure 4). This suggests that
*blt*/*blt* mature SP thymocytes have reduced
viability, at least in vitro, which is likely to contribute to the defect in vivo. The
failure of Bcl2 overexpression to rescue the peripheral T cell deficiency (Figure 4) suggests the involvement of cell-death
pathways other than those countered by Bcl2. We have revised the text to more clearly
discuss this point.

*4) In*
Figure 5*, the authors
used ChIP-Seq on total thymocytes to identify Zfp335 targets. Since the most relevant
cell type is mature thymocytes, they need to validate the target genes by performing
ChIP experiment using mature thymocytes followed by qPCR of the targets*.

As requested, we have performed ChIP-qPCR on sort-purified CD4SP thymocytes and
successfully validated the target genes in Figure 6, as well as reproduced the same pattern of differential Zfp335 binding at
various targets shown in Figure 6. These results
have now been added as Figure 6—figure supplement 1.

Reviewer 2:

*[…] Overall the analysis of the mutant mouse and gene demonstrated a critical
role of Zfp335 in development of T cells and identify the target genes, but failed to
neither reveal the mechanism of the developmental defect nor identify the function of
Zfp335 gene*.

*1) Although Zfp335 restored mature T cell development by bone marrow
reconstitution, it is recommended to confirm by the knock-in of Zfp335 because the
mice may still contain other mutations and the heterogeneity of chimeric mice does
not prove full restoration of development*.

While it is true that ENU mutagenesis gives rise to a scattering of mutations throughout
the genome, we have mentioned in the text that our whole-exome sequencing analysis
identified the Zfp335^R1092W^ mutation as the only novel homozygous
single-nucleotide variant within the mapped interval of interest on chromosome 2, making
it highly unlikely that the *bloto* phenotype is caused by some other
mutation. Given this information and the amount of data we have supporting the causative
role of the Zfp335^R1092W^ mutation, we hope that this reviewer will agree that
generating a knock-in mouse is not critical in the context of the present study. In the
longer term we agree that it will be valuable for comparisons to be made between the
phenotype of Zfp335^R1092W^ mice and mice lacking Zfp335 selectively in T
cells, and we make this point in the Discussion.

*2) Thymic selection using OTII-Tg mice showed significant defect of development
of mature TCR + T cells, the author might not simply neglect the effect on thymic
selection*.

As correctly pointed out and as stated in the text, OTII TCR-tg *blt/blt*
mice had reduced numbers of Vα2^+^ CD4SP thymocytes. However, the magnitude of
this decrease (approx. two-fold) was similar to what we see in non-TCR Tg
*blt/blt* mice (Figure 4 vs.
Figure 1), which suggests that the defect in
OTII mice probably represented the same maturation phenotype as that seen in polyclonal
mice. If there were a thymic selection effect on top of the maturation defect, we would
expect to see a greater fold reduction in OTII CD4SP thymocytes, which was not the case.
However, we agree that we cannot rule out a possible influence of
Zfp335^R1092W^ on thymocyte selection and we have revised the Discussion
section in an effort to clarify this point.

*3) The data for the effect of Ankle2 in the reconstitution experiment was
complicated such as the analysis of Rag-GFPhigh and low population (for example why
thy1.1 expression levels were so different between Ankle2 vs. control). Simpler
experiment and expression should be taken to show significant restoration from
developmental arrest*.

The difference in Thy1.1 reporter levels between Ankle2 and control is one that we
typically see in retroviral transduction experiments and may be explained by the fact
that the control cells were transduced with an empty retroviral vector, which due to its
smaller size compared to a vector containing a large gene like Ankle2 (2895 bp), is able
to integrate into the host genome more efficiently and be present at higher copy
numbers, resulting in a corresponding increase in reporter expression. We hope this
additional explanation helps clarify what is admittedly a complex experiment as we are
not aware of a simpler way to perform this analysis. We also note that the partial
rescue effect of Ankle2 was not observed with five other Zfp335 target genes tested in
reconstitution experiments (Figure 8—figure supplement 1), involving a cumulative analysis of more than 30 retrovirally transduced
*blt/blt* BM chimeric mice. We believe that this large comparison
group adds strength to the conclusion regarding the small but significant pro-maturation
effect of Ankle2.

Reviewer 3:

*[…] The data are convincing and well presented. In particular, the partial
rescue of the mutant phenotype by the overexpression of one of the identified Zfp335
targets provides strong evidence for the functional role of this Zfp335-regulated
gene. However, the study would gain additional significance by the analysis of mice
carrying a conditional null allele of Zfp335. Such mice have been published, and ES
cells that harbor a conditional null allele of Zfp335 are available from a mutant
mouse repository*.

*1) The authors need to address and/or discuss the discrepancies between the
previous and current studies of Zfp335. For example, in*
Figure 4—figure supplement 3
*the authors show that cell proliferation and survival are not affected by the
bloto mutation. However, the previous analysis of lymphoblastic cells of humans
carrying a hypomorphic H111R mutation shows an impaired growth of mutant
cells*.

We thank the reviewer for raising this point and have incorporated some of this
discussion into the revised text. The H1111R mutation described in human patients by
Yang et. al. was a far more severe hypomorph than the R1092W mutation present in our
mice. The human mutation affected splicing, resulting in lower levels of normally
spliced Zfp335 transcript and severely reduced protein expression in homozygous patient
lymphoblastic cell lines (16% of control). In contrast, the R1092W mutation was
comparatively benign: *blt*/*blt* cells have no decrease
in Zfp335 expression at the transcript or protein level. A plausible reason for the
discrepancy in proliferative capacity may be that H1111R mutant lymphoblastic cells
express very little functional Zfp335, while *blt*/*blt*
cells maintain sufficient Zfp335 activity such that they are capable of normal
proliferation and growth. We now discuss this possibility in the text.

*2) In the ChIP-seq analysis, shown in*
Figures 5 and 6*,
the authors used the same anti-Zfp335 antibodies as the previous study. However, the
authors identified a different sequence motif than the previously reported. Which
parameters were used in the MEME analysis? What are the other significantly enriched
motifs? Did the analysis detect the motif described in the Cell paper? If the motif
width range is adjusted, is the other motif detected? I appreciate the inclusion of
an electrophoretic mobility shift assay, which provides strong evidence for the newly
identified sequence motif. The authors should also include the previously reported
motif in this analysis*.

Using the set of parameters described in our Methods (min. width = 6, max. width = 30,
zero or one instance of a given motif per sequence) for MEME analysis of our thymocyte
ChIP-seq data, we did not detect the motif previously reported by Yang *et.
al.,* (Cell, 2012) within our top five highest ranked hits. Furthermore,
analysis of our data using an alternative motif discovery algorithm, HOMER, did not
reveal significant enrichment of this motif.

The MEME analysis parameters used by Yang *et. al.*, differed from ours
in two key aspects: 1) a maximum motif width of 20 (instead of 30) was specified; 2)
discriminative motif discovery was performed using a negative set of “background”
sequences contrasted against a set of target sequences extracted from the top 148 peaks
in their ChIP-seq dataset.

Even after adjusting for motif width, our analysis yielded the same sequence motif that
we identified, and not the previously reported motif. We have also re-analyzed the
embryonic brain ChIP-seq data, extracting 400 bp sequences from the top 148 peaks
identified by MACS, and running MEME with a maximum motif width of either 20 or 30. In
both cases, our proposed motif emerged as the top-scoring candidate, whereas the motif
reported by Yang et al., was not detected. Based on these analyses, we would argue that
the discrepancy between our conclusions and that of Yang *et. al.,* is
likely not due to true differences in the underlying biological data; instead,
differences in motif finding strategy, specifically with regards to the choice of a
background model by Yang *et. al.,* most likely account for this
discrepancy.

As suggested by the reviewer, we have performed additional experiments to test the
previously reported motif in an electrophoretic mobility shift assay (new Figure 7—figure supplement 1). The probe sequence
was derived from the Zfp335 binding site at the promoter of *Pdap1*, a
target gene identified by ChIP-seq in both embryonic brain and thymocyte ChIP-seq
datasets. Our assay did not reveal evidence for Zfp335 binding to this motif in vitro:
firstly, we were unable to detect formation of a gel-shift complex with labeled
*Pdap1* probe, and secondly, addition of excess unlabeled
*Pdap1* probe failed to compete against labeled Z1 probe (containing
our identified motif) for binding to Zfp335. These new data provide further support for
our newly identified Zfp335 recognition motif being a major target of Zfp335
binding.

*3) One of the strongest points of the paper is the partial rescue of the T cell
defect in bloto mice by the overexpression of Ankle2 (*Figure 8*). It would be of interest to
examine the phenotype of an Ankle2 knock-down*.

We agree, and in fact we had attempted to knock-down Ankle2 by transduction of bone
marrow progenitors using two different shRNA constructs (selected as the best of three
when tested in a cell line). Unfortunately, we were only able to achieve at most 50%
knock-down of Ankle2 expression in naïve T cells as assessed by qRT-PCR. At that level
of knock-down, there was no detectable effect on T cell maturation (unpublished data).
In *blt/blt* T cells, Ankle2 expression is decreased by a far greater
degree; at least ten-fold (Figure 8), which
suggests that our shRNA knock-down efficiency was insufficient to reveal any effects of
Ankle2 on T cell development. In the future, we hope to attempt this experiment again
using CRISPR to achieve more efficient knock-down.